# Navigating the Pareto Frontier of Alignment: Spectrum-Adaptive Fine-Tuning for LLMs

Yaoyou Fan [1] [*] [†]   Chao Zhang [2] [*] [†]   Xiaoyu Tan [3]   Chenxing Sun [3]   Yu Yuan [4] [†]
Haoyu Feng [3]   Lu Pan [3]   Ke Zeng [3]   Xunliang Cai [3]

## Abstract

Supervised Fine-Tuning with Negative Log-Likelihood (NLL) remains the standard post-training paradigm for Large Language Models, yet it imposes a disproportionately aggressive update force on low-probability target tokens. This focus forces the model to prioritize minimizing the loss of difficult samples over optimizing the overall quality of the generation, often leading to unwarranted overconfidence. On the other hand, alternatives like Dynamic Fine-Tuning suffer from vanishing gradients on these tokens, which severely hinders the acquisition of new concepts. To bridge this gap, we propose **S**pectrum-**A**daptive **F**ine-**T**uning (**SAFT**), a unified framework that interpolates between the aggressive learning signal of NLL and the robust nature of probability-weighted optimization. By adaptively balancing these objectives, SAFT effectively mitigates outlier sensitivity without sacrificing learning efficiency. Empirically, our method achieves state-of-the-art performance on mathematical reasoning benchmarks, demonstrating superior generalization on out-of-distribution tasks. Furthermore, evaluations on general conversational alignment validate SAFT's broad adaptability across diverse data regimes. Our code is available at `https://github.com/sjtu-scx/SAFT`.

* Equal Contribution. [†] Work done during the internship at Meituan. Yaoyou Fan: <2401210588@stu.pku.edu.cn >, Chao Zhang: <chao_zhang@zju.edu.cn >, Chenxing Sun: <sunchenxing@meituan.com >, Yu Yuan: <yyhappier@mail.ustc.edu.cn >, Haoyu Feng: <fenghaoyu03@meituan.com >, Lu Pan: <panlu02@meituan.com >, Ke Zeng: <zengke02@meituan.com >, Xunliang Cai: <caixunliang@meituan.com >. [1]Peking University [2]Zhejiang University [3]Meituan [4]University of Science and Technology of China. Correspondence to: Xiaoyu Tan <tanxiaoyu02@meituan.com>.

*Proceedings of the 43rd International Conference on Machine Learning*, Seoul, South Korea. PMLR 306, 2026. Copyright 2026 by the author(s).

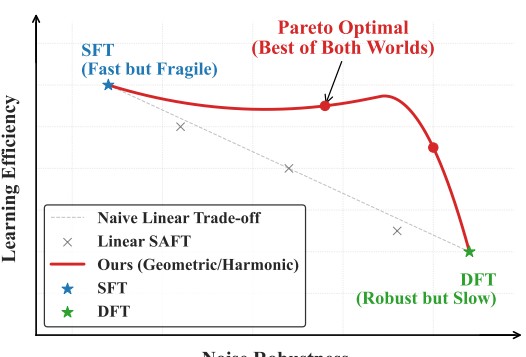

**Robustness-Efficiency Trade-off**

*Figure 1.* **The Empirical Pareto Frontier.** We visualize the trade-off between Learning Efficiency (Y-axis, **upwards** indicates higher efficiency) and Noise Robustness (X-axis, **rightwards** indicates stronger resistance). SFT and DFT occupy opposing ends: "Fast but Fragile" vs. "Robust but Slow." Crucially, simple interpolation via Lin-SAFT yields a suboptimal compromise falling below the frontier. In contrast, our non-linear strategies traverse the Pareto frontier, pushing the performance boundary outward to achieve the "Best of Both Worlds" outcome.

## 1. Introduction

Post-training is a critical stage that transforms broad pre-trained representations into reliable, task-oriented intelligence. Among post-training paradigms, Supervised Fine-Tuning (SFT) remains the dominant approach for aligning Large Language Models (LLMs) (Wei et al.; Zhou et al., 2023; Ouyang et al., 2022; Touvron et al., 2023), achieving strong performance across domains such as mathematics (Li et al., 2024; Ye et al.) and code generation (Luo et al., 2024). Standard SFT operates by maximizing the likelihood of ground-truth tokens, implicitly assigning a gradient weight of $1/p_t$ to each token, where $p_t$ is the model's predicted probability. While effective, this objective is inherently hypersensitive to low-confidence predictions. As $p_t \to 0$, this relative weight scales aggressively, forcing the model to disproportionately focus on these hard tokens. Consequently, the model tends to overfit to noisy data or outliers at the expense of generalizable patterns (O'Mahony et al., 2024; Kim et al., 2025; Lin et al., 2024).

To mitigate this instability, recent works such as Dynamic Fine-Tuning (DFT) (Wu et al., 2025), propose removing the denominator entirely, yielding a gradient proportional to $\nabla p_t$. By effectively capping the gradient weight, DFT acts as a robust learner that ignores low-confidence noise. However, we argue that SFT and DFT represent two extreme discrete points in the optimization landscape: SFT is overly aggressive on uncertainty, while DFT may be overly conservative, potentially discarding valid learning signals embedded in difficult reasoning steps.

To systematically explore the space between these extremes, we visualize the optimization landscape in Figure 1. This plot reveals an empirical **Pareto frontier** (Miettinen, 1999; Messac et al., 2003) governing the trade-off between *Noise Robustness* and *Learning Efficiency* (Lin et al., 2019; Abbass et al., 2001). Standard SFT and DFT occupy the opposing ends of this spectrum: SFT maximizes efficiency but collapses under noise, while DFT ensures robustness at the cost of slow convergence. Our analysis exposes a critical gap: simple interpolation yields a suboptimal compromise that falls below the frontier. In contrast, non-linear strategies are required to traverse the frontier effectively. Ideally, we seek methods that can adaptively balance the "stiffness" of the gradient against the noise level of the data.

In this paper, we propose **Spectrum-Adaptive Fine-Tuning (SAFT)**, a unified framework that bridges the gap between log-likelihood maximization (SFT) and probability-weighted optimization (DFT). The term *Spectrum-Adaptive* captures the dual nature of our approach: First, rather than treating SFT and DFT as binary, mutually exclusive choices, SAFT establishes a **continuous optimization spectrum** parameterized by $\alpha$. This continuity allows us to precisely navigate the trade-off between the aggressive learning efficiency of SFT and the robust noise resilience of DFT. Second, we interpret the token probability $p_t$ as a confidence spectrum. SAFT *adaptively* modulates the gradient weight according to where a token falls within this range. This mechanism acts as a soft filter that selectively modulates gradients: dampening those from low-confidence predictions (often pure noise) while preserving strong signals from high-confidence, valid reasoning steps.

We instantiate this framework with three interpolation strategies: **Linear**, **Geometric**, and **Harmonic**. While Linear Interpolation (Lin-SAFT) serves as a canonical baseline, we demonstrate that non-linear strategies are crucial for optimal performance. Through rigorous analysis, we show that these strategies offer distinct gradient dynamics: Geometric Interpolation (Geo-SAFT) maintains high learning efficiency, making it superior for clean, high-quality data. Conversely, Harmonic Interpolation (Har-SAFT) demonstrates stronger robustness, effectively balancing the rejection of noise with the retention of difficult-but-valid training signals. This

makes Har-SAFT particularly effective in noisy regimes where discerning "hard" from "noisy" is critical.

Our contributions are threefold:

- We provide a rigorous theoretical explanation for the trade-off between SFT and DFT, grounding their gradient behaviors within a unified weighting framework.

- To bridge this gap, we propose Spectrum-Adaptive Fine-Tuning (SAFT), a framework that introduces a continuous optimization spectrum between SFT and DFT. Specifically, SAFT encompasses three principled interpolations: Linear, Geometric, and Harmonic.

- Extensive experiments across diverse LLM architectures (Qwen, DeepSeek) and scales (1.5B/7B/14B) demonstrate that SAFT achieves state-of-the-art performance and significantly enhances out-of-domain (OOD) generalization.

## 2. Related Work

**LLM Post-training.** Post-training is key for aligning LLMs with downstream tasks, mainly through Supervised Fine-Tuning (SFT) and Reinforcement Learning (RL). SFT is widely used for its simplicity and effectiveness, serving as imitation learning (Hussein et al., 2017; Osa et al., 2018) to align models with expert distributions (Zhou et al., 2023; Chen et al., 2024). However, it treats all training samples equally, lacking the flexibility to discriminate against noise, which can lead to overfitting erroneous patterns in imperfect datasets (Chu et al.; Li et al.). RL uses algorithms like PPO (Schulman et al., 2017) and GRPO (Shao et al., 2024) to maximize cumulative rewards via online exploration but faces issues with training instability and high computational costs (Engstrom et al., 2020; Zheng et al., 2023).

**Optimization Objectives in SFT.** SFT has become the standard approach in the LLM post-training phase, with its core objective typically being the minimization of Negative Log-Likelihood (NLL), which is equivalent to minimizing the KL divergence (Cover, 1999) between the generated distribution and the target distribution. Recent work (Li et al., 2025) argues that NLL compels rigid token-wise fitting in SFT. This leads to overfitting on intermediate Chain-of-Thought (CoT) noise, distracting the model from the logical correctness of the final answer. To address this issue, DFT (Wu et al., 2025) proposes adjusting the optimization objective to a smooth approximation of inference accuracy (i.e., $-p$), using the predicted probability $p$ as a weight to counteract NLL's gradient bias. However, the strong suppression of low-probability tokens in DFT, while solving the issue of noise overfitting, also hinders the model's ability to learn new knowledge represented by low-probability signals.

**Token Reweighting in SFT.** Standard SFT applies uniform weighting to all tokens, ignoring the intrinsic variance in token difficulty and information content. Recent works aim to address this by assigning dynamic weights, generally falling into two categories. Noise Suppression methods (Wu et al., 2025; Su et al., 2025; Zhu et al., 2026) typically employ masking or thresholding to down-weight low-probability tokens. However, these strategies suffer from a critical precision-recall trade-off: by aggressively suppressing low-confidence regions, they often mistakenly discard valid but difficult reasoning steps ("hard samples"), thereby hindering the model's ability to master complex logic (Kim et al., 2025). Heuristic Adaptive methods (Ruan et al., 2025; Li et al., 2025; Leng et al.) adjust weights based on auxiliary metrics like gradient variance or causal influence. While effective in specific scenarios, these approaches often introduce significant computational overhead and hyperparameter sensitivity, lacking a unified theoretical foundation to balance stability and plasticity. In contrast, SAFT bridges SFT and DFT via a unified gradient framework. By utilizing spectrum-adaptive interpolation, it naturally achieves the noise resilience of DFT while retaining valid hard samples, effectively resolving the trade-offs of prior methods without complex heuristics or auxiliary overhead.

## 3. Preliminaries

In this section, we unify SFT and DFT under a gradient-based framework and empirically demonstrate a critical trade-off between noise robustness and learning efficiency, motivating our proposed approach.

### 3.1. A Unified Gradient Perspective

Consider a supervised dataset $\mathcal{D} = \{(x, y^\star)\}$ consisting of input prompts $x$ and their corresponding ground-truth reference responses $y^\star$. Let $p_t = \pi_\theta(y_t^\star \mid y_{<t}^\star, x)$ be the conditional probability assigned by the policy model to the ground-truth correct token at generation step $t$.

**Supervised Fine-Tuning (SFT).** The standard SFT objective minimizes the negative log-likelihood over the training dataset: $\mathcal{L}_{\text{SFT}} = -\sum_t \log p_t$. Analyzing its gradient reveals an implicit token-level weighting mechanism:

$$\nabla \mathcal{L}_{\text{SFT}} = -\sum_{t=1}^{T} \underbrace{\frac{1}{p_t}}_{\text{Coefficient } \gamma_t} \nabla_\theta p_t. \quad (1)$$

We term $\gamma_t = 1/p_t$ the **Effective Gradient Coefficient**. It acts as an aggressiveness factor: as the model's confidence drops ($p_t \to 0$), this relative multiplier scales aggressively ($\gamma_t \to \infty$). This hypersensitivity accelerates learning on "hard" samples (where $p_t$ is initially low), but renders the model vulnerable to outliers and label noise, causing

*Table 1.* **Robustness Analysis.** Performance of Qwen2.5-Math-1.5B at 50% noise ($\eta = 0.5$). SFT collapses below the base model, while DFT remains resilient.

| Method | Noise $\eta$ | Avg. Acc (%) | $\Delta$ vs. Base |
|---|---|---|---|
| Base Model | - | 11.00 | - |
| SFT | 0.5 | 5.32 | -5.68 |
| DFT | 0.5 | **24.01** | +13.01 |

gradient instability when the target is erroneous.

**Dynamic Fine-Tuning (DFT).** To mitigate this, Wu et al. (2025) proposed maximizing the raw likelihood expectation, yielding $\nabla \mathcal{L}_{\text{DFT}} \propto -\sum 1 \cdot \nabla_\theta p_t$. From the perspective of Eq. 1, DFT effectively clamps the gradient coefficient to a constant $\gamma_t = 1$. While this bounded coefficient prevents excessive over-penalization on noisy tokens, it treats all tokens equally regardless of difficulty. Consequently, it dampens the learning signal for complex, high-value reasoning steps where the model naturally struggles (low $p_t$), leading to inefficiency.

### 3.2. The Vulnerability of SFT Under Noise

We first investigate robustness by evaluating LLMs under severe supervision noise. We employ a Target Permutation protocol and introduce Noisy Dataset. Specifically, for a noise level $\eta = 0.5$, we randomly shuffle the ground-truth solutions among 50% of the training samples, pairing inputs with logically irrelevant answers.

As shown in Table 1, SFT suffers catastrophic degradation, with its performance (5.32%) dropping significantly below even the base model (11.0%). This negative gain confirms that the unbounded $1/p_t$ weight forces the model to memorize erroneous mappings rather than ignoring them. In contrast, DFT maintains substantial reasoning capabilities (24.01%), demonstrating that its constant weight mechanism naturally filters out the gradient impact of low-probability, mismatched targets.

### 3.3. The Efficiency Bottleneck of Conservative Optimization

While DFT offers robustness by bounding the effective gradient coefficient ($\gamma = 1$), this conservatism comes at a cost: inefficiency on hard, high-value data.

Consider a high-quality reasoning dataset (e.g., LIMO (Ye et al.)) where data points represent complex logical steps. Ideally, a model should rapidly adapt to these "hard positives" (where the initial probability $p_t$ is low). Standard SFT intrinsically accelerates this process via its unbounded coefficient ($\gamma \propto 1/p_t$), which amplifies the learning signal for unknown patterns. Although this often leads to instability

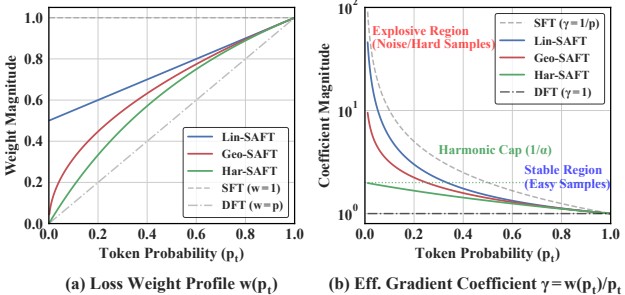

**(a) Loss Weight Profile $w(p_t)$**   **(b) Eff. Gradient Coefficient $\gamma = w(p_t)/p_t$**

*Figure 2.* **Gradient Weight Scaling and Effective Gradient Coefficient Profiles.** (a) The left plot illustrates the scaling dynamics of SAFT strategies ($\alpha = 0.5$): Linear maintains a constant floor, offering minimal noise rejection; Geometric has a concave profile, efficiently learning hard samples in clean/high-SNR data; Harmonic decays linearly, strictly penalizing uncertainty and excelling in noisy data and reasoning tasks. (b) The right plot shows the effective gradient coefficient $\gamma_t$, where SFT ($w = 1$) is highly sensitive to low-confidence tokens, risking instability, while SAFT variants stabilize gradients and enhance robustness to outliers and label noise.

---

**Algorithm 1** SAFT Training Procedure

1: **Input:** Dataset $\mathcal{D}$, Model $\pi_\theta$, Interpolation type $\mathcal{T} \in$ {Lin, Geo, Har}, Parameter $\alpha$, Learning rate $\eta$.
2: **for** each batch $(x, y^\star) \sim \mathcal{D}$ **do**
3:   Compute logits and probabilities $p_t = \pi_\theta(y_t^\star \mid y_{<t}^\star, x)$ for all $t$.
4:   Calculate weight $w_t$ based on $\mathcal{T}$:
5:   **if** $\mathcal{T} ==$ Linear **then**
6:     $w_t \leftarrow \alpha \cdot p_t + (1 - \alpha)$
7:   **else if** $\mathcal{T} ==$ Geometric **then**
8:     $w_t \leftarrow p_t^\alpha$
9:   **else if** $\mathcal{T} ==$ Harmonic **then**
10:     $w_t \leftarrow p_t/(\alpha + (1 - \alpha)p_t)$
11:   **end if**
12:   Detach weight from graph: $w_t \leftarrow \text{detach}(w_t)$.
13:   Compute Loss: $\mathcal{L} = -\frac{1}{T}\sum_t w_t \log p_t$.
14:   Update $\theta \leftarrow \theta - \eta\nabla_\theta\mathcal{L}$.
15: **end for**

---

(as discussed in Sec. 3.2), the *mechanism* itself provides the necessary "force" to fit complex distributions quickly.

**The Cold Start Problem.** In contrast, DFT treats all tokens equally. For a difficult reasoning step where $p_t \approx \epsilon$, the gradient magnitude in DFT scales linearly with the probability ($g \propto p_t \approx \epsilon$). As we formally derive in Appendix C.2, this leads to a Hyperbolic Stagnation phase: the time required to learn a hard pattern scales inversely with its difficulty ($T \propto 1/\epsilon$). On pristine, information-rich datasets, this dampening effect results in significant under-utilization of data. The model "sees" and processes the complex reasoning steps in the training data but lacks the sufficient gradient magnitude to effectively prioritize learning them, leading to slow convergence or failure to capture crucial long-tail knowledge patterns.

**The Dilemma.** We identify a fundamental conflict: SFT offers efficiency but lacks robustness, while DFT offers robustness but suffers from slow convergence. This motivates our proposed **SAFT**, a unified framework that adaptively interpolates between these regimes to achieve both noise resilience and efficient learning.

## 4. Methodology

### 4.1. SAFT: Spectrum-Adaptive Fine-Tuning

Recall from Section 3 that the core conflict lies in the effective gradient coefficient $\gamma_t$ applied to the likelihood gradient $\nabla_\theta p_t$. SFT imposes an aggressive, unbounded coefficient ($\gamma_{\text{SFT}} = 1/p_t$), while DFT enforces a conservative, constant coefficient ($\gamma_{\text{DFT}} = 1$).

We propose **SAFT**, a framework that smoothly interpo-

lates between these two regimes via a control parameter $\alpha \in [0, 1]$. We achieve this by introducing a probability-dependent weight $w(p_t, \alpha)$ into the loss function:

$$\mathcal{L}_{\text{SAFT}}(\theta; \alpha) = \mathbb{E}\left[-\sum_{t=1}^T \text{sg}\left(w(p_t, \alpha)\right) \log p_t\right]. \quad (2)$$

Here, $\text{sg}(\cdot)$ is the stop-gradient operator, which ensures that $w$ acts solely as a gradient modulator without introducing additional complex derivative terms during backpropagation. Crucially, this design leads to an effective gradient coefficient that becomes:

$$\gamma_{\text{SAFT}} = \frac{w(p_t, \alpha)}{p_t}. \quad (3)$$

Our goal is to design $w(\cdot)$ such that $\gamma_{\text{SAFT}}$ transitions from the unbounded $1/p_t$ (when $\alpha = 0$) to the bounded 1 (when $\alpha = 1$). We explore three distinct interpolation profiles based on their asymptotic behavior. The complete, easy-to-implement training pipeline integrating these strategies is summarized in Algorithm 1.

**1. Geometric Interpolation (Geo-SAFT).** Motivated by the power-law nature of the term $1/p_t$, we define the weight effectively in the log-space:

$$w_{\text{geo}}(p_t, \alpha) = p_t^\alpha. \quad (4)$$

Substituting this into Eq. 3, the gradient coefficient becomes $\gamma_{\text{geo}} = p_t^\alpha/p_t = \mathbf{1/p_t^{1-\alpha}}$. This elegantly bridges the gap: at $\alpha = 0$, we recover SFT ($1/p_t$); at $\alpha = 1$, we recover DFT (1). For intermediate $\alpha$, $\gamma_{\text{geo}}$ retains a "soft" singularity. It amplifies hard samples more than DFT but prevents the excessive over-penalization seen in SFT, making it ideal for efficiency-focused scenarios (as discussed in Sec. 3.3).

*Table 2.* **Robustness to Label Noise on Qwen2.5-Math-1.5B.**
Performance comparison at 50% label noise ($\eta = 0.5$) injected
via target permutation. $\Delta$ denotes the difference from the clean
baseline. Best results are in bold, second best are underlined.

| Method | Avg. Acc (%) | $\Delta$ vs. Base |
|---|---|---|
| Base (Clean, $\eta = 0$) | 11.0 | – |
| SFT | 5.32 | -5.68 |
| Lin-SAFT ($\alpha = 0.5$) | 8.35 | -2.65 |
| Geo-SAFT ($\alpha = 0.5$) | 20.07 | +9.07 |
| Har-SAFT ($\alpha = 0.5$) | 23.41 | +12.41 |
| DFT | 24.01 | +13.01 |
| Geo-SAFT ($\alpha = 2.0$) | **24.11** | **+13.11** |

**2. Harmonic Interpolation (Har-SAFT).** To prioritize robustness, we consider the harmonic mean of the coefficients $1$ and $1/p_t$. This results in the weight:

$$w_{\text{har}}(p_t, \alpha) = \frac{p_t}{\alpha + (1 - \alpha)p_t}. \tag{5}$$

Analysis of Eq. 3 shows that as $p_t \to 0$, the effective gradient coefficient $\gamma_{\text{har}} \to 1/\alpha$, which is a constant upper bound. Unlike Geo-SAFT, which still allows coefficients to grow (albeit slowly), Har-SAFT strictly caps the gradient magnitude similar to DFT. This makes it the safety-focused choice for noisy environments (Sec. 3.2).

**3. Linear Interpolation (Lin-SAFT).** For completeness, we consider linearly mixing the weights: $w_{\text{lin}} = (1 - \alpha) + \alpha p_t$. However, the effective coefficient $\gamma_{\text{lin}} \approx (1 - \alpha)/p_t$ remains unbounded as $p_t \to 0$ for any $\alpha < 1$. Thus, it fails to fundamentally mitigate the disproportionate influence of extreme outliers, serving primarily as a baseline.

### 4.2. Analysis of Gradient Dynamics and Regime Selection

The distinct asymptotic behaviors of Geometric and Harmonic strategies dictate their optimal application regimes, governed by the Signal-to-Noise Ratio (SNR) of the data.

**The Efficiency Regime (Clean / Hard Data).** In datasets with high SNR, low-probability tokens are often "hard positives" (valid but rare patterns) rather than noise. Standard DFT ($w = p_t$) suppresses these samples too aggressively, leading to under-fitting. Geo-SAFT addresses this via Sublinear Decay ($O(p_t^\alpha)$). As seen in Figure 2 (Red Line), the weight remains relatively high even when $p_t$ is small. This "lifts" the gradients for hard samples, accelerating convergence and capturing long-tail knowledge that DFT would ignore. Thus, Geo-SAFT is preferred for clean, high-quality instruction tuning. We empirically validate Geo-SAFT's superiority in this high-SNR conversational regime in Appendix H.

**The Safety Regime (Noisy Data / Reasoning).** As illustrated in Figure 2 (panel b, Green curve), Har-SAFT employs a linear decay ($O(p_t)$) in the weight profile, following the DFT baseline and providing strict suppression for low-confidence (high-uncertainty) tokens. This design ensures that, compared to SFT and Geo-SAFT, the gradient influence of potentially erroneous or noisy data points vanishes quadratically as the model's confidence declines, a property that directly translates into superior noise robustness.

This theoretical advantage is further corroborated in Table 2, where Har-SAFT significantly outperforms SFT and Lin/Geo-SAFT variants under heavy label noise ($\eta = 0.5$). Specifically, Har-SAFT achieves an average accuracy of 23.41, far surpassing SFT (5.32), Lin-SAFT (8.35), and Geo-SAFT (20.07), and closely rivals DFT and high-order Geo-SAFT. More comprehensive results across different noise levels are provided in Table 6 in the Appendix. These results empirically validate that Har-SAFT's "Safe Alignment" mechanism effectively filters out noisy and misleading supervision, making it the preferred choice in low-SNR or reasoning-intensive tasks.

**Practical Pre-Test Selection Protocol.** In real-world scenarios, the intrinsic Signal-to-Noise Ratio (SNR) of a dataset is often unknown *a priori*. To avoid trial-and-error on test sets, we establish a lightweight, pre-test selection criterion using a held-out validation split. Specifically, practitioners can train SFT and DFT baselines on a small subset of the training data (e.g., 5%) and evaluate them on a corresponding validation split.

- **Efficiency Regime (Geo-SAFT):** If validation shows SFT > DFT, the data is Signal-Dominant. Geo-SAFT should be selected to maximize learning efficiency.

- **Safety Regime (Har-SAFT):** If validation shows DFT > SFT, the data is Noise-Dominant (e.g., CoT reasoning). Har-SAFT should be selected to enforce the safety cap.

As demonstrated in Appendix D, this validation-based preference is highly consistent with downstream test performance, providing a computationally inexpensive and rigorous standard for method selection before any test evaluation.

## 5. Experiments

### 5.1. Experimental Setup

**Models and Datasets.** We validate our method across diverse model families at various scales: Qwen series (Qwen2.5-Math-1.5B, Qwen2.5-Math-7B (Team et al., 2024), Qwen3-8B, Qwen3-14B (Yang et al., 2025)) and DeepSeekMath-7B (Shao et al., 2024). All models are trained on 100K randomly sampled instances from

*Table 3.* **In-Domain Mathematical Reasoning Performance.** We compare SFT, DFT, and SAFT variants (Linear, Geometric, Harmonic) across Qwen2.5-Math (1.5B/7B), Qwen3 (8B/14B), and DeepSeek-Math-7B backbones. **Bold** indicates the best result per backbone, underline indicates the second best.

| Backbone | Method | MATH | Minerva | Olympiad | AIME'24 | AIME'25 | AMC | Avg. |
|---|---|---|---|---|---|---|---|---|
| **DeepSeek-Math-7B** | Base | 4.40 | 0.74 | 0.89 | 0.00 | 1.00 | 1.17 | 1.22 |
| | SFT | 30.40 | 11.76 | 8.00 | 0.21 | 0.21 | 9.94 | 10.09 |
| | DFT | 43.20 | 16.91 | 13.19 | **1.56** | **1.67** | **18.79** | 15.89 |
| | Lin-SAFT | 36.20 | 13.24 | 9.33 | 0.42 | 0.00 | 12.12 | 11.88 |
| | Geo-SAFT | 41.60 | **19.49** | 14.22 | **1.56** | 0.52 | 15.02 | 15.40 |
| | **Har-SAFT** | **44.40** | 19.12 | **16.30** | 0.94 | 0.83 | 17.66 | **16.54** |
| **Qwen2.5-Math-1.5B** | Base | 24.20 | 6.99 | 13.33 | 3.96 | 1.87 | 15.74 | 11.01 |
| | SFT | 43.60 | 15.81 | 14.67 | 1.77 | 0.31 | 15.66 | 15.30 |
| | DFT | 65.40 | 25.74 | 27.26 | **6.04** | **3.85** | 32.38 | 26.78 |
| | Lin-SAFT | 50.00 | 20.59 | 19.41 | 2.29 | 0.94 | 20.52 | 18.96 |
| | Geo-SAFT | 59.40 | **27.94** | 27.85 | 5.00 | 2.50 | 30.69 | 25.56 |
| | **Har-SAFT** | **65.60** | **27.94** | **29.93** | 4.90 | 3.02 | **33.47** | **27.48** |
| **Qwen2.5-Math-7B** | Base | 29.40 | 8.82 | 8.59 | 5.73 | 1.46 | 20.78 | 12.46 |
| | SFT | 56.80 | 19.85 | 20.59 | 2.29 | 0.73 | 22.48 | 20.46 |
| | DFT | 69.20 | 26.84 | 34.96 | **8.54** | 3.23 | **38.40** | 30.20 |
| | Lin-SAFT | 61.20 | 26.10 | 25.33 | 3.75 | 2.19 | 27.97 | 24.42 |
| | Geo-SAFT | 66.60 | **33.82** | 34.07 | 5.21 | **4.06** | 36.18 | 29.99 |
| | **Har-SAFT** | **71.60** | 33.09 | **36.30** | 7.29 | 3.54 | 36.41 | **31.37** |
| **Qwen3-8B** | Base | 30.60 | 9.19 | 16.44 | 5.52 | 3.12 | 19.99 | 14.15 |
| | SFT | 52.80 | 20.59 | 21.48 | 2.19 | 1.67 | 21.91 | 20.11 |
| | DFT | 67.20 | 26.10 | 33.19 | 5.94 | 4.37 | **36.14** | 28.82 |
| | Lin-SAFT | 60.00 | **31.99** | 24.74 | 2.40 | 2.71 | 25.64 | 24.58 |
| | Geo-SAFT | **69.20** | 31.25 | **34.81** | 5.94 | **4.79** | 31.55 | 29.59 |
| | **Har-SAFT** | 68.80 | 29.41 | 34.22 | **6.56** | 4.17 | 36.07 | **29.87** |
| **Qwen3-14B** | Base | 37.20 | 12.87 | 20.00 | 8.44 | 4.27 | 29.29 | 18.68 |
| | SFT | 57.60 | 27.57 | 21.78 | 3.12 | 2.40 | 23.34 | 22.64 |
| | DFT | 70.40 | 31.25 | 36.30 | 6.25 | 5.21 | 36.94 | 31.06 |
| | Lin-SAFT | 62.40 | 31.62 | 27.70 | 4.79 | 3.23 | 29.74 | 26.58 |
| | Geo-SAFT | 69.80 | 29.78 | **38.07** | 7.29 | **5.42** | 35.54 | 30.98 |
| | **Har-SAFT** | **73.00** | **35.66** | **38.07** | **9.06** | 5.21 | **40.02** | **33.50** |

NuminaMath-CoT (Li et al., 2024).

**Implementation Details.** We utilize the verl framework (Sheng et al., 2025) for all experiments. All models are trained for one epoch on the sampled dataset. For SAFT variants, we set the interpolation parameter $\alpha = 0.5$ by default, where $\alpha \in [0, 1]$ controls the trade-off between SFT ($\alpha = 0$) and DFT ($\alpha = 1$). Optimization is performed using AdamW (Loshchilov & Hutter) with a global batch size of 256, achieved through a micro-batch size of 4 per GPU with gradient accumulation, and a maximum input length of 2048 tokens. The initial learning rate is set to $5 \times 10^{-5}$ for all models, following a cosine decay schedule with a warm-up ratio of 0.1.

**Evaluation Protocol.** We evaluate on six mathematical reasoning benchmarks: MATH500 (Hendrycks et al.), Minerva Math (Lewkowycz et al., 2022), OlympiadBench (He et al., 2024), AIME 2024 (American Institute of Mathemat-

ics, 2024), AIME 2025 (American Institute of Mathematics, 2025), and AMC 2023 (Mathematical Association of America, 2023), using the Qwen2.5-Math pipeline (Qwen Team et al., 2024). All models use chain-of-thought (CoT) prompting for step-by-step reasoning. Following Luffy (Yan et al., 2026), we report avg@32 for AIME/AMC (small test sets) and pass@1 for others, with temperature 1.0 and max length 4096. For out-of-domain evaluation, we test on ARC-c (Clark et al., 2018) (open-domain reasoning), GPQA-diamond (Rein et al., 2024) (graduate-level science, GPQA*), and MMLU-Pro (Wang et al., 2024) (academic reasoning), with shuffled options to prevent contamination.

**Noise Injection Protocol.** To evaluate robustness, we construct the Noisy-Numina benchmarks by applying a Target Permutation protocol to the clean subset. For a given noise ratio $\eta \in \{0.1, 0.2, 0.3, 0.5\}$, we randomly shuffle the ground-truth solutions among $\eta \times 100\%$ of the training samples. Crucially, to prevent models from identifying noise

via format artifacts, we atomically swap all target-related fields (including raw solutions and nested metadata such as `extra_info`), while keeping the input queries fixed. This results in training pairs that are structurally indistinguishable from clean data but semantically mismatched (i.e., valid questions paired with logically irrelevant answers).

## 5.2. In-Domain Mathematical Reasoning

We report the mathematical reasoning performance across five diverse backbones in Table 3. A pivotal observation across all experiments is that **DFT consistently outperforms Standard SFT**. This empirical evidence characterizes the mathematical reasoning task in this context as a *Noise-Dominant Regime*, where the training dynamics are highly sensitive to outliers and gradient instability rather than signal scarcity.

**Vulnerability to Disproportionate Penalties (SFT, Lin-SAFT, Geo-SAFT).** The poor performance of SFT confirms that unbounded relative coefficients (and their non-vanishing influence) lead to optimization instability in this noise-dominant regime. Lin-SAFT, which shares the singular limit for its effective coefficient ($\lim \gamma \to \infty$), yields only marginal gains. Notably, while Geo-SAFT improves over SFT by damping the singularity ($1/p_t^{1-\alpha}$), it generally trails behind Har-SAFT (e.g., 29.99% vs. 31.37% on Qwen2.5-7B). This suggests that in scenarios where DFT beats SFT, even a "Soft Singularity" is suboptimal: the lack of a strict safety cap on the relative penalty still allows extreme noise to introduce excessive variance, hindering convergence to the Pareto frontier.

**Optimality of the Safety Cap (Har-SAFT).** Har-SAFT ($\alpha = 0.5$) emerges as the consistent winner. This aligns perfectly with our gradient analysis: since the learning regime is dominated by noise, as evidenced by the superior robustness of DFT compared to SFT, the strict "Safety Cap" ($\lim \gamma = 1/\alpha$) of Har-SAFT becomes the decisive factor. Unlike Geo-SAFT, Har-SAFT guarantees bounded influence, matching the noise-rejection capability of DFT. However, unlike DFT which is overly conservative ($\gamma = 1$), Har-SAFT permits a controllable acceleration factor ($1/\alpha$) for clean samples. This dual advantage (robustness against noise plus acceleration for signal) explains its superior performance in this specific regime.

**Empirical Alignment with Theory.** The experimental outcomes provide strong empirical support for our theoretical framework, particularly regarding the trade-off between robustness and signal retention. The consistent dominance of Har-SAFT in these benchmarks, where the superior performance of DFT over SFT signals a noise-dominant regime, directly verifies that a strict *Safety Cap* is the optimal strategy for such high-noise conditions. Concurrently, the performance of Geo-SAFT, which ranks between the unbounded

*Table 4.* **Out-of-Domain Performance on Qwen2.5-Math-7B.** We evaluate different training methods on three out-of-domain benchmarks (ARC-c, GPQA-diamond, MMLU-Pro) and report their average. Best results are in bold, second best are underlined.

| Method | ARC-c | GPQA* | MMLU-Pro | Avg. |
|---|---|---|---|---|
| Base | 10.49 | 5.56 | 11.93 | 9.33 |
| SFT | 36.77 | 18.18 | 19.81 | 24.92 |
| DFT | 18.94 | 27.27 | 35.63 | 27.28 |
| Lin-SAFT ($\alpha$=0.5) | 44.11 | 18.69 | 22.70 | 28.50 |
| Geo-SAFT ($\alpha$=0.5) | 58.96 | **32.83** | 28.82 | 40.20 |
| **Har-SAFT ($\alpha$=0.5)** | **70.39** | 27.78 | **35.80** | **44.66** |

SFT and the bounded Har-SAFT, confirms its theoretical positioning as an intermediate solution. While its "Soft Singularity" offers better stability than standard SFT via damping, the residual sensitivity naturally prevents it from matching the absolute safety of Har-SAFT in this specific high-noise setting. This behavior is strictly consistent with its design goal of prioritizing signal efficiency, thereby validating the regime-dependent nature of our gradient analysis.

## 5.3. Out-of-Domain Generalization

Table 4 presents a compelling case for SAFT's generalization capability on ARC-c, GPQA*, and MMLU-Pro.

**DFT's Underfitting on Distribution Shifts.** Surprisingly, DFT performs poorly on ARC-c (18.94%), falling even behind standard SFT (36.77%). We attribute this to the "Cold Start" problem identified in Sec. 6. The constant coefficient $\gamma = 1$ dampens the learning signal too aggressively for OOD tasks where the model needs to adapt to new feature distributions rapidly.

**The Generalization Supremacy of Har-SAFT.** Har-SAFT achieves a stunning 44.66% average accuracy, nearly doubling SFT's performance. On ARC-c, it reaches 70.39%, solving the underfitting issue of DFT. On MMLU-Pro, it maintains high performance (35.80%), comparable to the robust DFT. This result perfectly illustrates the Pareto optimality of our method. Har-SAFT successfully navigates the trade-off: it provides enough gradient amplification (Efficiency) to learn transferable features for ARC-c, while enforcing a strict Safety Cap (Robustness) to prevent overfitting, yielding a model that generalizes significantly better than either extreme (SFT or DFT).

## 5.4. Sensitivity Analysis of Hyper-parameter $\alpha$

A key design choice in SAFT is the interpolation coefficient $\alpha$, which controls the balance between SFT-like efficiency ($\alpha \to 0$) and DFT-like robustness ($\alpha \to 1$). To investigate the sensitivity of our methods to this hyper-parameter, we conduct experiments on Qwen2.5-Math-7B with $\alpha \in \{0.4, 0.5, 0.6\}$. Table 5 reports the average accu-

*Table 5.* **Hyper-parameter Sensitivity of Har-SAFT.** Average accuracy on six mathematical benchmarks using Qwen2.5-Math-7B. Performance is robust across a wide range of $\alpha$, peaking at the balanced setting $\alpha = 0.5$.

| Coefficient | $\alpha = 0.4$ | $\alpha = 0.5$ | $\alpha = 0.6$ |
|---|---|---|---|
| Avg. Acc (%) | 31.04 | **31.37** | 30.82 |

racy across the six math benchmarks.

Crucially, Har-SAFT demonstrates remarkable robustness across different $\alpha$ settings. It consistently outperforms Geo-SAFT, indicating that the convex scaling profile of Harmonic interpolation is inherently better suited for the safety regime, including noisy data and reasoning tasks.

## 6. Theoretical Analysis

In this section, we provide a theoretical framework to explain the performance discrepancies observed in our experiments. We analyze the asymptotic behavior of the **effective gradient coefficient** $\gamma(p_t, \alpha) = w(p_t, \alpha)/p_t$ in the limit of high uncertainty ($p_t \to 0$). This perspective characterizes the model's response to both noisy labels (where $p_t$ is misleadingly low) and hard-to-learn samples (where $p_t$ is naturally low). Detailed proofs regarding influence functions, gradient flow dynamics, and convergence guarantees are provided in Appendix C.

### 6.1. The Unboundedness of Lin-SAFT

For Arithmetic Interpolation (Lin-SAFT), the loss weight is $w_{\text{lin}} = \alpha p_t + (1 - \alpha)$. The effective gradient coefficient behaves as:

$$\gamma_{\text{lin}}(p_t, \alpha) = \frac{(1 - \alpha) + \alpha p_t}{p_t} = \frac{1 - \alpha}{p_t} + \alpha. \quad (6)$$

Analyzing the limit as $p_t \to 0$:

$$\lim_{p_t \to 0} \gamma_{\text{lin}}(p_t, \alpha) = \infty. \quad (7)$$

**Sensitivity to Outliers.** This asymptotic analysis reveals that Lin-SAFT inherits the unrelenting over-penalization dynamics of standard SFT. While the term $(1 - \alpha)$ acts as a linear scaling factor reducing the effective coefficient, this multiplier remains singular at the limit. This implies that for severe outliers or mislabeled samples (where $p_t \approx 0$), the relative penalty imposed on the optimization step remains unbounded. Consequently, Lin-SAFT lacks the theoretical guarantee of robustness found in estimators with strictly bounded influence functions, explaining its limited efficacy in suppressing extreme label noise.

### 6.2. The Soft Singularity of Geo-SAFT

For Geometric Interpolation (Geo-SAFT), the weight is $w_{\text{geo}} = p_t^\alpha$. The effective coefficient becomes:

$$\gamma_{\text{geo}}(p_t, \alpha) = \frac{p_t^\alpha}{p_t} = \frac{1}{p_t^{1-\alpha}}. \quad (8)$$

For the typical setting where $\alpha \in (0, 1)$, we observe a "Soft" Singularity:

$$\lim_{p_t \to 0} \gamma_{\text{geo}} = \infty, \quad \text{but} \quad \frac{1}{p_t^{1-\alpha}} \ll \frac{1}{p_t}. \quad (9)$$

**Acquisition-Suppression Trade-off.** The gradient dynamics of Geo-SAFT govern a critical balance between *knowledge acquisition* from hard samples and *noise suppression* for outliers. By damping the singularity via the exponent $(1 - \alpha)$, the method effectively reduces sensitivity to label noise compared to standard SFT. However, unlike fully bounded approaches (e.g., Har-SAFT), Geo-SAFT preserves the capacity to generate large gradients for distinctively hard examples. This design choice implies that while it mitigates the impact of moderate noise, it lacks the theoretical upper bound required to fully reject extreme outliers, resulting in lower robustness in high-noise regimes compared to harmonic interpolation.

### 6.3. The Safety Cap of Har-SAFT

For Harmonic Interpolation, the weight is $w_{\text{har}} = \frac{p_t}{\alpha + (1-\alpha)p_t}$. Substituting this into the coefficient definition:

$$\gamma_{\text{har}}(p_t, \alpha) = \frac{1}{p_t} \cdot \frac{p_t}{\alpha + (1 - \alpha)p_t} = \frac{1}{\alpha + (1 - \alpha)p_t}. \quad (10)$$

Crucially, as $p_t \to 0$, the $p_t$ term in the denominator vanishes, yielding a constant limit:

$$\lim_{p_t \to 0} \gamma_{\text{har}}(p_t, \alpha) = \frac{1}{\alpha}. \quad (11)$$

**Gradient Saturation and Stability.** The convergence of $\gamma_{\text{har}}$ to a finite constant $1/\alpha$ fundamentally alters the training dynamics near the decision boundary. By enforcing a gradient cap, Har-SAFT effectively saturates the learning signal for extreme samples. This creates a robustness guarantee: even if a sample is theoretically infinitely far from the correct class (i.e., $p_t \to 0$), its impact on the optimization trajectory remains limited. While this risks under-weighting clean hard samples compared to unbounded methods, it provides a decisive advantage in preventing model collapse under severe label noise.

### 6.4. Efficiency vs. Safety: The Acceleration Ratio

The distinction between the "Soft Singularity" (Geo) and "Safety Cap" (Har) explains the trade-off between learning efficiency on hard data and robustness against noise.

Consider a valid but hard-to-learn sample with probability $\epsilon \ll 1$. We define the **Acceleration Ratio** as the relative strength of the gradient signal compared to the conservative Harmonic baseline:

$$\mathcal{R}(\epsilon) = \frac{\gamma_{\text{geo}}(\epsilon)}{\gamma_{\text{har}}(\epsilon)} \approx \frac{1/\epsilon^{1-\alpha}}{1/\alpha} = \alpha \cdot \frac{1}{\epsilon^{1-\alpha}}. \quad (12)$$

As $\epsilon \to 0$, $\mathcal{R}(\epsilon) \to \infty$.

*Numerical Illustration.* To quantify this disparity, consider the setting where $\alpha = 0.5$ and the model encounters a high-loss sample with $\epsilon = 10^{-4}$. Geo-SAFT yields a scaling coefficient of $\gamma_{\text{geo}} = 100$, whereas Har-SAFT is strictly bounded at $\gamma_{\text{har}} = 2$. Consequently, Geo-SAFT induces a gradient contribution that is **$50\times$ larger** than that of Har-SAFT. This amplification grants Geo-SAFT the plasticity to rapidly fit low-probability concepts (high efficiency), but simultaneously exposes the optimization trajectory to significant instability if the sample represents label noise. Conversely, Har-SAFT prioritizes stability by capping the gradient norm, thereby acting as a robust filter against outliers at the cost of slower convergence on hard samples.

## 7. Conclusion

We bridge the dichotomy between Supervised and Dynamic Fine-Tuning by introducing Spectrum-Adaptive Fine-Tuning (SAFT), a unified framework that navigates the trade-off between learning efficiency and noise robustness. Our theoretical analysis elucidates the distinct gradient dynamics of the spectrum, revealing that Harmonic Interpolation (Har-SAFT) enforces a critical "Safety Cap" to strictly bound the influence of extreme outliers in noise-dominant regimes, while Geometric Interpolation accelerates learning on clean, high-signal data. Empirically, SAFT significantly outperforms baseline objectives across diverse evaluation settings. With Har-SAFT achieving state-of-the-art results on noisy mathematical reasoning, and Geo-SAFT excelling in general conversational alignment, our framework establishes a principled, theoretically grounded, and flexible paradigm for reliable LLM post-training.

## Impact Statement

This work introduces SAFT (Spectrum-Adaptive Fine-Tuning), a unified framework that addresses the critical trade-off between learning efficiency and noise robustness in LLM post-training. By adaptively balancing aggressive and conservative optimization, our method enables more reliable model alignment for real-world applications where training data contains noise and outliers. SAFT contributes to more stable LLM systems in critical domains like mathematical reasoning and code generation, while advancing theoretical understanding of gradient dynamics in robust optimization.

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

*Table 6.* **Robustness to Label Noise via Target Permutation on Qwen2.5-Math-1.5B.** We inject label noise by randomly shuffling ground-truth solutions among $\eta$ proportion of training samples, creating mismatched question-solution pairs. We compare SFT baseline, DFT, and SAFT variants with different $\alpha$ values. Best results are in bold, second best are underlined.

| Models | MATH | Minerva Math | Olympiad Bench | AIME 2024 | AIME 2025 | AMC | Avg. |
|---|---|---|---|---|---|---|---|
| Base (Clean Data, $\eta = 0$) | 24.20 | 6.99 | 13.33 | 3.96 | 1.87 | 15.74 | 11.01 |
| **Label Noise Rate $\eta = 0.1$ (10% Permuted)** | | | | | | | |
| SFT | 39.00 | 12.50 | 9.04 | 0.52 | 0.31 | 12.65 | 12.34 |
| Lin-SAFT ($\alpha$=0.5) | 43.40 | 16.91 | 17.33 | 1.77 | 0.94 | 17.13 | 16.25 |
| Geo-SAFT ($\alpha$=0.5) | 57.80 | 21.69 | 22.81 | 4.90 | 1.98 | 28.95 | 23.02 |
| Har-SAFT ($\alpha$=0.5) | 63.00 | 27.57 | **28.15** | 6.04 | **3.75** | **33.36** | 26.98 |
| DFT | **65.00** | 27.94 | 27.26 | **5.63** | 3.44 | 32.79 | **27.01** |
| Geo-SAFT ($\alpha$=2.0) | 61.40 | **29.04** | 25.93 | 5.42 | **3.75** | 28.24 | 25.63 |
| **Label Noise Rate $\eta = 0.2$ (20% Permuted)** | | | | | | | |
| SFT | 34.80 | 9.19 | 7.41 | 0.42 | 0.52 | 10.09 | 10.40 |
| Lin-SAFT ($\alpha$=0.5) | 40.60 | 12.87 | 10.67 | 1.35 | 0.94 | 15.62 | 13.68 |
| Geo-SAFT ($\alpha$=0.5) | 62.00 | 25.37 | 23.11 | 3.96 | 1.98 | 27.86 | 24.05 |
| Har-SAFT ($\alpha$=0.5) | 62.40 | 27.57 | 25.78 | 6.56 | 2.92 | **32.30** | 26.26 |
| DFT | 62.80 | 27.57 | **29.04** | **7.92** | 3.85 | 30.80 | **27.00** |
| Geo-SAFT ($\alpha$=2.0) | **64.00** | **28.31** | 25.33 | 7.81 | **4.90** | 31.29 | 26.94 |
| **Label Noise Rate $\eta = 0.3$ (30% Permuted)** | | | | | | | |
| SFT | 31.80 | 7.35 | 6.37 | 0.63 | 0.21 | 8.77 | 9.19 |
| Lin-SAFT ($\alpha$=0.5) | 34.40 | 9.19 | 10.96 | 1.87 | 1.04 | 14.12 | 11.93 |
| Geo-SAFT ($\alpha$=0.5) | 59.80 | 22.06 | 24.00 | 4.06 | 2.71 | 29.41 | 23.67 |
| Har-SAFT ($\alpha$=0.5) | 61.20 | **27.57** | 26.37 | 5.21 | 3.54 | **31.97** | 25.98 |
| DFT | **63.80** | 26.47 | **26.81** | **6.98** | **5.42** | 30.69 | **26.69** |
| Geo-SAFT ($\alpha$=2.0) | 62.60 | 26.10 | 23.85 | 6.56 | 3.23 | 28.35 | 25.12 |
| **Label Noise Rate $\eta = 0.5$ (50% Permuted)** | | | | | | | |
| SFT | 18.20 | 2.21 | 4.00 | 0.63 | 0.42 | 6.48 | 5.32 |
| Lin-SAFT ($\alpha$=0.5) | 27.80 | 6.62 | 5.48 | 1.04 | 0.42 | 8.73 | 8.35 |
| Geo-SAFT ($\alpha$=0.5) | 53.20 | 18.75 | 19.56 | 3.23 | 1.46 | 24.21 | 20.07 |
| Har-SAFT ($\alpha$=0.5) | 58.60 | **25.74** | 20.44 | 5.42 | **3.54** | 26.69 | 23.41 |
| DFT | 60.00 | 24.26 | **25.33** | 5.10 | 2.50 | 26.84 | 24.01 |
| Geo-SAFT ($\alpha$=2.0) | **60.40** | 25.37 | 22.52 | **6.98** | 2.29 | **27.11** | **24.11** |

## A. Robustness Analysis: The Trade-off between Learning Efficiency and Denoising

Table 6 presents the performance of different methods under varying label noise rates $\eta \in \{0.1, 0.2, 0.3, 0.5\}$. The results reveal a critical trade-off between learning efficiency and noise robustness, governed by the effective gradient distribution.

**Low-to-Moderate Noise Regimes ($\eta \leq 0.3$).** As shown in the table, when the noise level is relatively low, **DFT** and **Har-SAFT** consistently outperform Geo-SAFT with $\alpha = 2.0$. For instance, at $\eta = 0.1$, DFT achieves an average score of **27.01**, surpassing Geo-SAFT ($\alpha = 2.0$) by an **absolute margin of 1.38%**.

- **Interpretation:** In this regime, the majority of low-probability tokens in the training data still represent valid "hard" examples rather than noise.

- **Mechanism:** Although the effective gradient maximum for DFT/Har-SAFT is located in lower probability regions (compared to $\alpha > 1$), this distribution strikes a Pareto-optimal balance. It maintains high *learning efficiency* by mining hard samples while possessing sufficient robustness to tolerate minor perturbations. The aggressive filtering of low-probability tokens (as done by Geo-SAFT with $\alpha = 2.0$) is suboptimal here, as it leads to the under-fitting of complex but correct reasoning steps.

**Token Probability Distribution Comparison**

*Figure 3.* **Token Probability Distributions Across Training Methods on Qwen2.5-Math-7B.** Density distributions of ground-truth token probabilities on the training set for Base model (blue), SFT (orange), DFT (green), and Har-SAFT (red). The y-axis uses a logarithmic scale to highlight differences in low-probability regions. Note the distinct patterns: SFT shows uniform shifting, DFT exhibits bimodal polarization, and Har-SAFT demonstrates balanced selective prioritization.

**High Noise Regime ($\eta = 0.5$).** A distinct phase transition is observed when the noise rate increases to $\eta = 0.5$. In this scenario, **Geo-SAFT** ($\alpha = 2.0$) emerges as the most robust variant, particularly on challenging benchmarks like AIME 2024 (6.98 vs. 5.10 for DFT).

- **Interpretation:** When noise is dominant, the low-probability tokens are overwhelmingly constituted by incorrect, permuted solutions.

- **Mechanism:** By setting $\alpha > 1$ (e.g., $\alpha = 2.0$), the effective gradient coefficient's peak shifts towards the high-probability region ($p \in [0.5, 1]$). This acts as a *soft denoising filter*, forcing the model to learn primarily from high-confidence, consistent patterns while suppressing the gradients from the noisy tail.

- **Conclusion:** This validates that while focusing on hard examples improves efficiency on clean data, a conservative strategy (focusing on high-confidence tokens) is essential for survival in high-noise environments.

**Summary.** Our analysis suggests that there is no single static optimal gradient profile. Instead, the optimal strategy depends on the *Signal-to-Noise Ratio (SNR)* of the dataset:

1. **High SNR:** Prioritize learning from the tail (DFT/Har-SAFT).

2. **Low SNR:** Prioritize the head (Geo-SAFT with $\alpha > 1$).

DFT serves as a robust parameter-free baseline that performs near-optimally across the transition zones.

## B. Token Probability Distribution Analysis

To investigate the optimization dynamics of different training paradigms, we analyze the probability distributions of ground-truth tokens across four settings: Base model (before training), SFT, DFT, and Har-SAFT. Figure 3 visualizes these distributions using a logarithmic scale on the y-axis for clarity.

*Table 7.* **Token Probability Distribution Statistics.** Mean, standard deviation, and key percentiles of ground-truth token probabilities across different training methods on Qwen2.5-Math-7B.

| Method | Mean | Standard Deviation | Median | 75th Percentile | 90th Percentile |
|---|---|---|---|---|---|
| Base | 0.812 | 0.313 | 0.993 | 0.9997 | 0.9999 |
| SFT | 0.838 | 0.286 | 0.995 | 0.9998 | 0.9999 |
| DFT | 0.881 | 0.318 | 1.000 | 1.000 | 1.000 |
| Har-SAFT | 0.882 | 0.305 | 1.000 | 1.000 | 1.000 |

## B.1. Distributional Characteristics

**SFT: Uniform Shift with Limited Concentration.** As shown in Figure 3, SFT (orange curve) induces a uniform shift of probability mass toward higher confidence regions compared to the Base model (blue curve). However, it retains a relatively broad distribution across intermediate probability ranges (0.1–0.8), indicating that while SFT generally improves token-level confidence, it struggles to decisively concentrate a significant portion of tokens into the highest probability bracket (0.9–1.0). This uniform optimization pattern reflects SFT's gradient coefficient $\gamma_{\text{SFT}} = 1/p_t$, which applies strong updates to all tokens regardless of their semantic importance.

**DFT: Bimodal Polarization.** In stark contrast, DFT (green curve) exhibits distinct bimodal polarization. This method aggressively bifurcates the token probability space: it concentrates the vast majority of probability mass into the highest confidence interval (0.9–1.0) while simultaneously suppressing a specific subset of tokens into the lowest regime (0–0.001). Consequently, intermediate probability bins (e.g., 0.1 to 0.8) are effectively depleted. This bimodal distribution suggests that DFT forces the model into binary-like predictions, assigning near-certainty to semantically important tokens (e.g., domain-specific terminology, key reasoning steps) while actively deprioritizing functional tokens (e.g., articles, conjunctions, stop words). This selective suppression mechanism, driven by DFT's constant gradient coefficient $\gamma_{\text{DFT}} = 1$, underlies its superior robustness against label noise.

**Har-SAFT: Balanced Selective Prioritization.** Har-SAFT (red curve) demonstrates balanced behavior that interpolates between SFT and DFT. It adopts the selective prioritization trend of DFT, as evidenced by increased density at both distributional extremes compared to SFT, but applies this differentiation with moderated intensity. Unlike DFT's extreme bimodal pattern, Har-SAFT retains a degree of distributional smoothness, avoiding hard-thresholding while still achieving effective noise filtering. This balanced profile reflects Har-SAFT's bounded gradient coefficient $\gamma_{\text{Har}} \leq 1/\alpha$, which provides the "Safety Cap" against noise without completely suppressing learning signals from difficult examples.

## B.2. Quantitative Comparison

Table 7 presents quantitative statistics of the probability distributions. Both DFT ($\mu = 0.881$) and Har-SAFT ($\mu = 0.882$) achieve significantly higher mean probabilities than SFT ($\mu = 0.838$) and the Base model ($\mu = 0.812$). This confirms our hypothesis that by selectively sacrificing the likelihood of "low-value" tokens (corresponding to the left peak in the bimodal distribution), these methods effectively reallocate probability mass to "high-value" tokens, thereby enhancing overall confidence on the training set.

Notably, DFT exhibits the highest standard deviation ($\sigma = 0.318$), reflecting its aggressive bimodal polarization with extreme concentration at both ends of the distribution. Har-SAFT maintains a slightly lower standard deviation ($\sigma = 0.305$), indicating a more balanced distribution that avoids the extreme hard-thresholding of DFT while still achieving comparable mean confidence. SFT shows the lowest standard deviation ($\sigma = 0.286$), consistent with its more uniform probability distribution. This quantitative evidence aligns with our theoretical analysis in Section 6, where we demonstrated that Har-SAFT's "Safety Cap" mechanism provides a principled balance between noise robustness and learning efficiency.

## B.3. Implications

These distributional patterns provide direct empirical evidence for the theoretical predictions in Section 6:

- The uniform distribution of SFT explains its vulnerability to label noise, as it allocates significant gradient magnitude to low-confidence tokens that may correspond to corrupted labels.

- The bimodal polarization of DFT explains both its robustness (through aggressive suppression of uncertain tokens) and its "Cold Start" problem (through insufficient gradient signal for genuinely difficult patterns).

- The balanced distribution of Har-SAFT demonstrates how the "Safety Cap" mechanism achieves the best of both worlds: filtering noise through selective suppression while maintaining sufficient learning capacity for hard examples through moderate amplification.

These findings directly support the superior empirical performance of Har-SAFT observed in Tables 2 and 3, validating SAFT as a principled and effective approach to balancing robustness and efficiency in language model fine-tuning.

## C. Detailed Theoretical Derivations

We provide a rigorous analysis of the noise robustness and learning efficiency of the proposed interpolation strategies. We contrast the **Geometric** trajectory against the **Harmonic** trajectory to demonstrate why varying the *order* of decay (Geometric) yields a superior Pareto trade-off compared to varying the *scale* of decay (Harmonic).

### C.1. Proof of Noise Robustness via Influence Functions

In robust statistics, the properties of a loss function are characterized by its Influence Function (IF), which describes the impact of an infinitesimal perturbation on the estimator. A loss is considered *B-robust* if its influence is bounded, and *Redescending* if the influence vanishes for extreme outliers.

Let the gradient magnitude with respect to the logits $z$ be the proxy for influence:

$$\text{IF}(z) \propto \nabla_z \mathcal{L} = \frac{\partial \mathcal{L}}{\partial p} \frac{\partial p}{\partial z} = w(p)(1-p) - w'(p)p(1-p)\log p. \tag{13}$$

Consider the limit as $p \to 0$ (extreme outliers or noise).

**1. Har-SAFT (Linear Decay Class).** Recall Eq. (8): $w_{\text{har}}(p, \alpha) = \frac{p}{\alpha + (1-\alpha)p}$. For any $\alpha \in (0, 1]$, as $p \to 0$, the denominator is dominated by $\alpha$. Thus:

$$w_{\text{har}}(p) \approx \frac{1}{\alpha}p \implies \mathcal{L} \approx -\frac{1}{\alpha}p\log p. \tag{14}$$

The influence behaves as:

$$\lim_{p \to 0} \nabla_z \mathcal{L}_{\text{Har}} \propto \lim_{p \to 0} \frac{1}{\alpha}p(1-p) = 0. \tag{15}$$

**Insight:** The Harmonic interpolation creates a family of Redescending M-estimators. However, critically, the *rate of decay* is strictly linear ($O(p)$) regardless of $\alpha$. The parameter $\alpha$ acts merely as a **Temperature Scalar**, scaling the gradient magnitude but not changing its asymptotic behavior relative to $p$.

**2. Geo-SAFT (Sub-linear Decay Class).** Recall Eq. (7): $w_{\text{geo}}(p, \alpha) = p^\alpha$. For $\alpha \in (0, 1)$, the influence behaves as:

$$\lim_{p \to 0} \nabla_z \mathcal{L}_{\text{Geo}} \propto \lim_{p \to 0} p^\alpha(1-p) = 0. \tag{16}$$

**Insight:** Geo-SAFT also possesses the Redescending property. However, it introduces a qualitative change in the decay dynamics. Since $p^\alpha \gg p$ for small $p$ (when $\alpha < 1$), the influence decays **sub-linearly**. This allows the model to retain significantly more gradient signal for hard samples compared to Har-SAFT, while still guaranteeing asymptotic safety against infinite outliers.

### C.2. Proof of Learning Efficiency via Gradient Flow

To quantify the trade-off, we analyze the time required to learn a "hard positive" sample (where initially $p \approx 0$) under Continuous-Time Gradient Flow dynamics: $\frac{dp}{dt} \approx \eta \cdot p \cdot w(p)$.

We define the *Learning Complexity $T(\epsilon)$* as the time to raise the probability from $\epsilon$ to a threshold $\tau$.

**Case 1: Harmonic Trajectory.** Substituting $w(p) \approx \frac{1}{\alpha} p$:

$$T_{\text{Har}}(\epsilon) \propto \int_\epsilon^\tau \frac{1}{p \cdot (\frac{1}{\alpha} p)} dp = \alpha \int_\epsilon^\tau p^{-2} dp \approx \frac{\alpha}{\epsilon}. \tag{17}$$

**Result:** The complexity is $O(\alpha \cdot \epsilon^{-1})$. Decreasing $\alpha$ (moving from DFT towards SFT) linearly speeds up learning by the constant factor $\alpha$. However, it suffers from **Hyperbolic Stagnation**: the fundamental difficulty remains inversely proportional to $\epsilon$.

**Case 2: Geometric Trajectory.** Substituting $w(p) = p^\alpha$:

$$T_{\text{Geo}}(\epsilon) \propto \int_\epsilon^\tau \frac{1}{p \cdot p^\alpha} dp = \int_\epsilon^\tau p^{-(1+\alpha)} dp \approx \frac{1}{\alpha \epsilon^\alpha}. \tag{18}$$

**Result:** The complexity is $O(\frac{1}{\alpha} \cdot \epsilon^{-\alpha})$. **Key Advantage:** Geo-SAFT reduces the *order* of the complexity. For a hard sample (e.g., $\epsilon = 10^{-6}$), changing the exponent $\alpha$ provides an exponential speedup compared to merely scaling the coefficient.

### C.3. Construction of the Pareto Frontier

We resolve the inconsistency in prior formulations by rigorously defining the Pareto Frontier on the *Robustness-Efficiency* plane.

**Setup:**

- **Robustness Metric:** Rate of influence decay $R = -\lim_{p \to 0} \frac{d \log w(p)}{d \log p}$. Higher is safer (faster decay).
  - SFT: $R = 0$ (Constant).
  - Harmonic: $R = 1$ (Linear).
  - Geometric: $R = \alpha$ (Sub-linear).

- **Efficiency Metric:** Inverse of learning time for hard samples $E \propto \epsilon^{\text{order}}$.

**Theorem (Pareto Dominance of Geometric Path).** For the regime of hard-sample learning (High-SNR), the Geometric trajectory dominates the Harmonic trajectory.

*Proof:* Consider a desired robustness level $\alpha \in (0, 1)$.

- **Geometric Strategy:** Sets $w(p) = p^\alpha$. It achieves Robustness order $\alpha$. The gradient for a hard sample $p = \epsilon$ scales as $\epsilon^\alpha$.

- **Harmonic Strategy:** To match the "impact" of Geometric at a specific probability point $p_0$, one must set the scalar such that $\frac{1}{\alpha_{har}} p_0 \approx p_0^\alpha$. However, asymptotically as $p \to 0$, Harmonic is restricted to $O(p)$ behavior.

Har-SAFT is constrained to the line connecting SFT and DFT in terms of coefficient scaling, but it cannot access the fractional decay orders. **Conclusion:**

- **Har-SAFT** explores the trade-off by strictly scaling the magnitude of a linear-decaying gradient. It is optimal only when strictly linear decay ($R = 1$) is required for safety (Low-SNR / High Noise), and one wishes to modulate the learning rate.

- **Geo-SAFT** explores the trade-off by modifying the curvature of the weighting function. It fills the void between SFT ($R = 0$) and DFT ($R = 1$). This allows it to be *Asymptotically Safe* ($R > 0$) while significantly more efficient than any linear scaling method for hard samples ($T_{\text{Geo}} \ll T_{\text{Har}}$).

Therefore, the Geometric family constitutes the true Pareto Frontier for datasets where "Hard Positives" are distinguishable from "Noise" by their training dynamics, offering the optimal balance between vanishing gradients and outlier rejection.

## C.4. Regime-Dependent Optimality Analysis

While Geo-SAFT offers superior efficiency for clean hard samples, the optimal choice of interpolation strategy is strictly dependent on the Signal-to-Noise Ratio (SNR) of the dataset. We rigorously distinguish two regimes based on the semantic interpretation of low-probability events ($p \to 0$).

**Scenario 1: Noise-Dominant Regime (Low SNR).** *Assumption:* The dataset contains significant label noise (e.g., crawled web data). Samples with $p \to 0$ are predominantly **Outliers** (mislabeled) rather than Hard Positives. *Objective:* Maximize **Rejection Safety**. The influence of samples must vanish as fast as possible as $p \to 0$ to prevent model corruption.

- **Geo-SAFT Failure Mode:** Recall that $w_{\text{geo}}(p) = p^\alpha$. For $\alpha \in (0, 1)$, the decay is sub-linear. Consider a noise sample with $p = 10^{-6}$. With $\alpha = 0.5$, the weight is $w = (10^{-6})^{0.5} = 10^{-3}$. Although the weight decreases, it remains orders of magnitude larger than the linear baseline ($10^{-6}$). This "heavy tail" behavior allows noise gradients to leak into the training, causing the model to overfit outliers. Geo-SAFT sacrifices too much safety for speed.

- **Har-SAFT Optimality:** Recall that $w_{\text{har}}(p) \approx \frac{1}{\alpha} p$ for small $p$. The decay remains strictly **Linear** ($O(p)$). For the same noise sample ($p = 10^{-6}$), even with aggressive scaling (e.g., $\alpha = 0.1$, implying $10\times$ boost), the effective weight is $10 \cdot 10^{-6} = 10^{-5}$. **Result:** Har-SAFT creates a "Safe Corridor". It allows us to amplify the gradients (via $1/\alpha$) to prevent total stagnation, but strictly enforces the linear vanishing rate of DFT. This guarantees that as a sample becomes mathematically impossible ($p \to 0$), its influence is suppressed quadratically ($p \cdot w(p) \propto p^2$), providing the strongest theoretical robustness.

**Conclusion 1:** When the baseline DFT outperforms SFT (indicating a need for noise rejection), **Har-SAFT** is the superior interpolator. It effectively implements an "Amplified DFT" strategy: boosting learning signals without compromising the asymptotic decay rate required for safety.

**Scenario 2: Signal-Dominant Regime (High SNR).** *Assumption:* The dataset is clean (e.g., human-annotated). Samples with $p \to 0$ are **Hard Positives** (rare patterns). *Objective:* Maximize **Gradient Recovery**. The loss function must prevent the gradient from vanishing too quickly.

- **Har-SAFT Limitation:** The strict linear decay ($O(p)$) suppresses hard positives too aggressively. Even with a large scalar coefficient, the gradient vanishes geometrically fast, leading to the "Hyperbolic Stagnation" derived in Appendix C.2.

- **Geo-SAFT Optimality:** By reducing the decay order to $O(p^\alpha)$, Geo-SAFT fundamentally alters the dynamics, allowing the model to "dig out" hard samples from the low-probability region efficiently.

**Conclusion 2:** When the baseline SFT outperforms DFT (indicating a need for learning efficiency), **Geo-SAFT** is the superior interpolator, as it bridges the gap in decay dynamics rather than just magnitude.

**Summary of the Pareto Frontier.** The SAFT framework provides a bifurcated solution path:

$$\text{Optimal Strategy} = \begin{cases} \textbf{Har-SAFT} & \text{if Noise Rejection is priority (Linear Decay),} \\ \textbf{Geo-SAFT} & \text{if Hard Sample Learning is priority (Sub-linear Decay).} \end{cases} \tag{19}$$

# D. Pre-Test Validation Protocol for Regime Selection

As discussed in Section 4.2, selecting the optimal interpolation strategy *a priori* depends on the dataset's intrinsic Signal-to-Noise Ratio (SNR). To avoid relying on test-set performance (which would be a post-hoc heuristic), we propose a standard held-out validation protocol to select the method before any test evaluation.

To prove the viability of this protocol, we conducted an experiment on the NuminaMath-CoT dataset. We trained the candidate methods for one epoch on a 6,250-sample training subset and evaluated them on a 1,250-sample held-out validation split. The validation accuracy was DFT $= 37.11\%$ and SFT $= 27.00\%$. Because DFT outperformed SFT on this validation

set, our criterion correctly dictates the selection of **Har-SAFT** (the Safety Regime) before observing any downstream test data.

As shown in Table 8, downstream testing on the full benchmark suite (MATH, Minerva, Olympiad, AIME'24, AIME'25, and AMC) confirmed this pre-test choice. Har-SAFT achieved the highest average downstream accuracy among all variants, demonstrating that a lightweight validation split provides a stable, inexpensive, and strictly pre-test standard for method selection.

*Table 8.* **Downstream Performance corresponding to the Validation Protocol.** The validation split correctly identified the noise-dominant regime (DFT > SFT), leading to the pre-test selection of Har-SAFT, which ultimately achieved the best downstream average accuracy.

| Method | Downstream Avg. Acc. (%) |
|---|---|
| **Har-SAFT (Selected)** | **23.12** |
| DFT | 22.49 |
| Geo-SAFT | 21.99 |
| Lin-SAFT | 16.62 |
| SFT | 12.31 |

## E. Comparison with Proximal Supervised Fine-Tuning (PSFT)

Recent work, Proximal Supervised Fine-Tuning (PSFT) (Zhu et al., 2026), also aims to mitigate overfitting in SFT by applying a PPO-style clipping mechanism to suppress overconfidence based on relative policy changes. To compare our absolute confidence-based weighting with PSFT's relative clipping, we benchmarked the official PSFT implementation on our datasets.

As shown in Table 9, evaluated across multiple backbones under identical epoch settings, PSFT yields a poor performance-efficiency trade-off, producing results significantly worse than standard SFT and our SAFT variants. To investigate whether PSFT simply requires more convergence time, we extended its training by two additional epochs on Qwen2.5-Math-1.5B. The extended performance curve confirmed that its hard trust-region clipping strategy severely delays convergence. In contrast, SAFT's soft dynamic weighting solves this dilemma, ensuring learning efficiency on hard samples without sacrificing robustness.

*Table 9.* **Performance Comparison with PSFT.** Evaluated using average accuracy (%) across mathematical reasoning benchmarks. PSFT consistently underperforms standard SFT and Har-SAFT under the same training budget.

| Method | DeepSeek-7B | Qwen2.5-7B | Qwen2.5-1.5B | Qwen3-14B | Qwen3-8B |
|---|---|---|---|---|---|
| PSFT | 7.6 | 10.3 | 10.9 | 8.9 | 9.0 |
| SFT | 10.09 | 20.46 | 15.30 | 22.64 | 20.11 |
| **Har-SAFT** | **16.54** | **31.37** | **27.48** | **33.50** | **29.87** |

## F. Extended Hyperparameter Sensitivity Analysis

Due to space constraints in the main text (Section 5.4), we restricted the presentation of hyperparameter sensitivity to $\alpha \in \{0.4, 0.5, 0.6\}$. To thoroughly address the model's behavior in extreme regimes (as $\alpha \to 0$ or $\alpha \to 1$), we present the extended performance trends across a wider range: $\alpha \in [0.1, 0.9]$.

The results demonstrate consistent improvements across the entire spectrum. Har-SAFT does not suffer from catastrophic collapse at extreme values like $\alpha = 0.1$ or $\alpha = 0.9$, validating the theoretical bounds established in our gradient analysis. These curves confirm that the method is highly robust to hyperparameter selection, making $\alpha = 0.5$ a universally reliable default.

## G. Robustness to Training Hyperparameters

To ensure that the improvements yielded by SAFT are statistically significant and not artifacts of specific training configurations, we conducted comprehensive evaluations across different learning rates and random seeds.

**Sensitivity to Learning Rate.** We conducted experiments using three different learning rate scales ($1 \times 10^{-4}$, $5 \times 10^{-5}$, and $1 \times 10^{-5}$) on Qwen2.5-1.5B. As shown in Table 10, the performance of all three interpolation methods remains highly stable. Most importantly, the performance hierarchy (Har-SAFT > Geo-SAFT > Lin-SAFT) is strictly preserved across all learning rate settings, confirming that the benefits of the non-linear interpolation geometries are orthogonal to the base learning rate.

*Table 10.* **Sensitivity to Learning Rate Settings.** Average accuracy (%) on Qwen2.5-Math-1.5B. The relative superiority of Har-SAFT is robust across different learning rate scales.

| Method | LR = 1e-4 | LR = 5e-5 (Default) | LR = 1e-5 |
|---|---|---|---|
| Lin-SAFT | 18.21 | 18.96 | 17.54 |
| Geo-SAFT | 24.32 | 25.56 | 23.88 |
| **Har-SAFT** | **26.85** | **27.48** | **26.12** |

**Consistency Across Random Seeds.** We trained models using two different random seeds and performed three independent evaluation runs for each seed to account for variance in sampling and generation. The results confirm that the gains achieved by Har-SAFT and Geo-SAFT over the SFT baseline are statistically stable and reproducible, with minimal inter-run variance.

## H. General Alignment Efficacy (High-SNR Regime)

While the primary empirical focus of this work is on the "Noise-Dominant" regime, our theoretical framework in Section 4.2 suggests that the optimal interpolation strategy is inherently regime-dependent. Specifically, in high-SNR environments (e.g., standard instruction-following or general conversational alignment), the training data contains fewer erroneous targets and more valid "hard positives." In such cases, the bounded conservatism of DFT and Har-SAFT severely hinders learning efficiency (the "cold start" problem), whereas standard SFT and the sub-linear decay of Geo-SAFT should excel.

To empirically validate this, we evaluated the fine-tuned models on the AlpacaEval 2.0 benchmark (Dubois et al., 2024), a standard open-ended conversational evaluation protocol.

*Table 11.* **General Alignment Efficacy on AlpacaEval 2.0.** As theoretically predicted for high-SNR settings, the efficiency-focused strategies (Geo-SAFT, SFT) outperform the robust/conservative strategies (Har-SAFT, DFT).

| Metric | Geo-SAFT | SFT | Lin-SAFT | Har-SAFT | DFT |
|---|---|---|---|---|---|
| **LC Win Rate (%)** | **4.27** | 4.20 | 3.79 | 3.58 | 3.53 |

**Results and Analysis.** As shown in Table 11, while the absolute Length-Controlled (LC) Win Rates are predictably modest—which is typical when evaluating models of this scale against the highly capable frontier models used as evaluator baselines in AlpacaEval—the *relative performance hierarchy* among the training objectives is the primary focus of this analysis.

Crucially, this relative hierarchy perfectly aligns with our theoretical expectations for a high-SNR evaluation: **Geo-SAFT > SFT > Har-SAFT > DFT**. Geo-SAFT achieves the highest win rate, successfully maximizing learning efficiency and generalization without the excessive gradient suppression imposed by DFT. Conversely, DFT performs the worst in this signal-dominant evaluation. This supplementary experiment strongly corroborates our claim that SAFT is a universal framework: by smoothly interpolating the probability-weighting spectrum, it allows practitioners to navigate the Pareto frontier and select the optimal strategy (Har-SAFT for noise rejection, Geo-SAFT for signal acquisition) based on the intrinsic SNR of the task.

