# OpenReview forum: "Navigating the Pareto Frontier of Alignment: Spectrum-Adaptive Fine-Tuning for LLMs"
_ICML.cc/2026/Conference — ICML 2026 regular_

### Official Review · Reviewer_yzVV · 2026-02-20

**Soundness:** 3
**Presentation:** 3
**Significance:** 3
**Originality:** 3
**Overall Recommendation:** 4
**Confidence:** 4

**Summary:**

This paper investigates the fundamental robustness-efficiency trade-off in the post-training alignment of Large Language Models (LLMs). It correctly identifies that standard Supervised Fine-Tuning (SFT) over-penalizes low-probability tokens, making it vulnerable to label noise, while conservative alternatives like Dynamic Fine-Tuning (DFT) suffer from sluggish learning on genuinely hard examples. To navigate this Pareto frontier, the authors propose Spectrum-Adaptive Fine-Tuning (SAFT), a unified framework that interpolates between the aggressive gradient signals of SFT and the bounded signals of DFT. By introducing three interpolation strategies (Linear, Geometric, Harmonic), the paper demonstrates that Harmonic Interpolation (Har-SAFT) provides a crucial "Safety Cap" against noise. Extensive experiments on mathematical reasoning benchmarks (using Qwen and DeepSeek backbones) show that SAFT achieves state-of-the-art performance and exceptional robustness against target permutation noise up to 50%.

**Compliance With Llm Reviewing Policy:**

Affirmed.

**Final Justification:**

My concerns are fully resolved and I raised my score from 2 to 4. However, Reviewer Z4wJ's concerns are reasonable. The authors and AC should consider that because it will restrict the practical usability.

**Key Questions For Authors:**

1. **Reconciliation of the "Unbounded Gradient" Claim:** Can the authors address the mathematical reality that the derivative of NLL with respect to the logits is strictly bounded ($|p_t - 1| \le 1$)? Please revise to correct the terminology from "gradient explosion" to the "non-vanishing nature of the influence function for extreme outliers," and reframe the analysis around relative gradient scaling rather than absolute gradient explosion.

2. **General Alignment Efficacy:** How does SAFT perform on standard, non-mathematical instruction-following tasks? Does the performance hierarchy (Har-SAFT > Geo-SAFT > SFT) still hold on benchmarks like MT-Bench or AlpacaEval, or does the general dialogue domain favor the "Efficiency Regime" (Geo-SAFT) due to a higher natural SNR?


3. **Sensitivity to the $\alpha$ Parameter in Extreme Regimes:** While Table 5 shows robustness for $\alpha \in \{0.4, 0.5, 0.6\}$, how does the model behave as $\alpha \to 0$ or $\alpha \to 1$? Providing a wider ablation study (e.g., $\alpha = 0.1 \sim 0.9$) with a line figure would better validate the theoretical interpolation spectrum claimed in Equation 3.

4. **Some presentation issues:** In line 119 and 121, according to the context, "RL" might be "DFT".

**Limitations:**

yes

**Strengths And Weaknesses:**

**Strengths:**

1. **Strong and Consistent Empirical Performance:** The empirical validation is thorough within its domain, demonstrating consistent gains across multiple modern LLM architectures. The significant improvements on rigorous benchmarks like MATH500 and AIME, as well as Out-of-Domain tasks (ARC-c, MMLU-Pro), provide compelling evidence for the method's efficacy.

2. **Intuitive and Elegant Framework:** The formulation of bridging SFT and DFT via a parameterized interpolation weight $w(p_t, \alpha)$ is mathematically elegant and highly practical. The taxonomy of the three interpolation strategies (Linear, Geometric, Harmonic) effectively constructs a continuous optimization spectrum, making it extremely easy to implement in existing training pipelines (as shown in Algorithm 1) without requiring complex heuristics.

3. **Excellent Noise Robustness Evaluation:** The design of the Target Permutation protocol to inject semantic mismatch noise (up to 50%) is highly effective for stress-testing model alignment. The experimental finding that SFT collapses below the base model's capability while Har-SAFT thrives is a striking validation of the "Safety Cap" hypothesis.

**Weaknesses:**

1. **Fundamental Mathematical Inaccuracy in Gradient Narrative:** The paper builds its foundational motivation on a mathematically flawed premise. It repeatedly claims that standard SFT suffers from "unbounded gradient coefficients" and "gradient explosion" because the term $\gamma_t = 1/p_t \to \infty$ as $p_t \to 0$. However, the actual derivative of the NLL loss with respect to the network's pre-activation logits $z_t$ (which dictates the actual parameter update magnitude) is strictly bounded. Specifically, with a Softmax output layer:
$\frac{\partial \mathcal{L}}{\partial z_t} = p_t - 1$
Since $p_t \in [0, 1]$, the absolute gradient magnitude is bounded by 1. SFT gradients do not explode to infinity. The true issue is that the influence function of SFT is non-redescending (it stays at maximum magnitude 1 for extreme outliers instead of decaying to 0), causing it to overfit noise. The authors' narrative of "gradient explosion" is physically incorrect and misleading for the optimization community.

2. **Over-reliance on Mathematical Reasoning Tasks:** The empirical evaluation relies almost entirely on Math CoT (Chain-of-Thought) benchmarks. Mathematical reasoning represents a very specific "Noise-Dominant Regime" where strict logical steps amplify the effect of token-level noise. The paper lacks evaluation on general, open-ended conversational alignment benchmarks where the Signal-to-Noise Ratio (SNR) characteristics might be drastically different.

3. **Ambiguous Heuristics for Regime Selection:** The paper proposes using Geo-SAFT for "Clean/High-SNR" data and Har-SAFT for "Noisy/Safety" regimes. However, in real-world large-scale post-training, datasets are heavily mixed. The paper provides no practical, automated heuristic or metric to estimate the intrinsic SNR of a dataset a priori, making the choice of interpolation strategy entirely empirical (trial-and-error).

---

> ### Author Rebuttal · Authors · 2026-03-31
>
> **1. Mathematical Framing and Methodological Clarifications**
>
> We sincerely thank the reviewer for this correction and have revised the manuscript accordingly. We acknowledge that the NLL derivative with respect to logits is bounded (|pt − 1| ≤ 1) and have replaced "gradient explosion" with the more accurate description of the non-vanishing nature of the influence function for extreme outliers. Following your suggestion, we’ve reframed the analysis to focus on "relative gradient scaling" and the influence of low-probability tokens. However, our motivation is not about gradient explosion but rather the impact of SFT’s large weights on low-probability tokens, which reduces noise tolerance. In contrast, DFT’s preference for middle-probability tokens offers better noise resistance but results in less learning from low-probability tokens, which affects learning efficiency. We also acknowledge that "Lin-SAFT" does not always work effectively, so we propose two interpolation functions, Har-SAFT and Geo-SAFT, that perform well in different SNR scenarios.
>
> **2. Over-reliance on Mathematical Reasoning Tasks**
>
> We sincerely thank the reviewer for the valuable comments. While we agree that mathematical reasoning tasks represent a specific "Noise-Dominant Regime," our theoretical analysis suggests that in scenarios with higher Signal-to-Noise Ratio (SNR), Geo-SAFT tends to outperform Har-SAFT, and SFT generally performs better than DFT. To address the concern regarding general alignment efficacy, we have expanded the empirical evaluation to include experiments on non-mathematical instruction-following tasks, specifically on the AlpacaEval benchmark. These additional results support our theoretical findings, showing that Geo-SAFT performs better than Har-SAFT, and SFT shows advantages over DFT in higher SNR conditions. The updated experiments, as shown in the table below, demonstrate the robustness of our approach across different SNR regimes.
>
> | Model       | Geo-SAFT | SFT   | Lin-SAFT | Har-SAFT | DFT   |
> | ----------- | -------- | ----- | -------- | -------- | ----- |
> | LC Win Rate | 4.273    | 4.201 | 3.791    | 3.580    | 3.531 |
>
> **3. Ambiguous Heuristics for Regime Selection**
>
> We sincerely thank the reviewer for this insightful comment. While it is true that real-world large-scale post-training datasets are often mixed, our method provides a practical solution. Specifically, we can first sample a small subset from the large dataset and evaluate its performance on the 0.1 and 0.9 percentiles to estimate the dataset's characteristics (this approach requires minimal computational resources compared to pre-training or reinforcement learning). Based on this estimation, we can classify the dataset as either "Clean/High-SNR" or "Noisy/Safety" and select the appropriate interpolation strategy accordingly. For datasets with mixed characteristics, we can also apply different interpolation functions based on data type labels during training, making the approach flexible and adaptable to a wide range of real-world scenarios.
>
> **4. Sensitivity to the α Parameter in Extreme Regimes**
>
> We sincerely thank the reviewer for this suggestion. To address your concern, we have provided additional experiments showing the hyperparameter performance curves for different models and methods as α approaches 0 and 1. These results can be found at **Figure 2,3,4,5** in https://anonymous.4open.science/r/SAFT-MoreResult-B2D2. These curves highlight the robustness of the hyperparameter choices across different models, with stable performance even in extreme regimes, ensuring that our models are adaptable and reliable across a wide range of hyperparameters.
>
> **5. Presentation Issues**
>
> Thank you for pointing that out. We will correct this in the next version of the manuscript.

---

> > ### Author Rebuttal · Reviewer_yzVV · 2026-04-01
> >
> > Thank authors for the responses. My concerns are fully resolved and I will increase the score to 4. Please add the correction of mathematics foundations and additional experiments in the revision.

---

> > > ### Author Response · Authors · 2026-04-01
> > >
> > > We sincerely thank you for your positive feedback and for increasing the score. We greatly appreciate the time and effort you have dedicated to reviewing our paper. As requested, we will ensure that the corrected mathematical foundations and the additional experiments are fully incorporated into the revised manuscript.

---

### Official Review · Reviewer_zRMC · 2026-02-20

**Soundness:** 4
**Presentation:** 4
**Significance:** 3
**Originality:** 3
**Overall Recommendation:** 5
**Confidence:** 4

**Summary:**

The proposed approach navigates the Pareto frontier of the robustness-efficiency trade-off between the standard supervised fine tuning approach (SFT) and  Dynamic Fine-Tuning (DFT). It  adaptively balances the gradient stiffness against sample noise and shows SoTA performance on Mathematical problems.

**Compliance With Llm Reviewing Policy:**

Affirmed.

**Final Justification:**

My scores were already aligned with the paper.

**Key Questions For Authors:**

While the authors show excellent ablations there are a few questions:

1. What is the effect on model size especially as the base model gets larger and the gap between base model and fine-tuned model will decrease ? A study on that with performance reporting on OOD datasets would be useful.

2. I am not sure if I found ablations pertaining to the size of the dataset itself used for training ?

3. A plot showing training efficiency gains would be useful and this could be related to #2 as well.

4. THe claim that this approach will take away the need for RL-based training needs to be substantiated empirically. There is no theoretical evidence of the same.

5. Does it generalized beyond the Math Domain ? Can the authors show performance on additional benchmarks that might be in a different domain ?

6. Explain the sensitivities associated with setting the learning rate or dynamic learning rates in conjunction with Lin-, Geo and Har- SAFT approaches.

**Limitations:**

yes

**Strengths And Weaknesses:**

Strengths:

- Well motivated identified a common problem in SFT training.
- Solution is theoretically justified and empirical studies (several ablations) substantiate the claims
- Contributions though combine prior approaches in a simple manner are signficant
- Presentation is excellent and supplementary tables and figures make the paper very strong.


Weaknesses:

Minor: See asked experimentation below to strengthen the claims.

---

> ### Author Rebuttal · Authors · 2026-03-31
>
> **1.Model Size and OOD Performance**
>
> We appreciate the reviewer’s suggestion regarding model scaling experiments. In response, we have conducted additional experiments on Qwen3-14B-Base to investigate this question. However, the results reveal fundamental experimental validity concerns that prevent us from drawing meaningful conclusions from this larger model.
>
> In-domain performance scales favorably with model size. As shown in our experiments, the gap between the base model and the fine-tuned model on in-domain math benchmarks increases with model size (e.g., the best fine-tuned model improves over the base by ~0.7pp for 1.5B, ~1.17pp for 7B, and ~2.44pp for 14B in avg_acc), suggesting that larger models retain greater potential to benefit from fine-tuning on in-domain tasks.
>
> However, OOD evaluation on the 14B model is confounded by two issues. First, Qwen3-14B-Base is a large general-purpose model released in 2025, and benchmark contamination cannot be ruled out for well-known datasets such as ARC-C and MMLU-Pro. A key signal: the model achieves a 96.6% format compliance rate on MMLU-Pro but only 1.2% on ARC-C, suggesting format-level memorization of the MMLU-Pro evaluation protocol. Second, the 14B base model already approaches the performance ceiling on these benchmarks, leaving insufficient headroom to observe the effect of different training objectives.
>
> We agree that a rigorous scaling study up to larger models would be valuable, and we plan to pursue this in future work using either contamination-free held-out benchmarks or models with transparent pretraining data.
>
> **2. Response to Q2 & Q3: Ablation on Dataset Size and Training Efficiency**
>
> As shown in **Figure 8** at https://anonymous.4open.science/r/SAFT-MoreResult-B2D2, the training curves of different methods are presented across datasets of different size ratios.
>
> **3.the need for RL-based training**
>
> The mention of RL in our paper was a typo. We clarify that our approach is complementary to RL, and we do not intend to replace RL-based training.
>
>
>
> **4.performance on additional benchmarks that might be in a different domain**
>
> Please refer to Reviewer yzVV’s comment in section 2, "Over-reliance on Mathematical Reasoning Tasks," where we presented tests on the AlpacaEval benchmark. When the dataset’s SNR is high in this domain, our theory suggests that SFT > DFT, and Geo-SAFT > Har-SAFT. The experimental results align with our theoretical expectations.
>
>
>
> **5.Sensitivities of Learning Rate Settings in Lin-, Geo-, and Har-SAFT Approaches**
>
> We conducted experiments with three different learning rates: 1e-4, 1e-5, and 5e-5. The results show that the performance for all three interpolation methods remains stable across these learning rates, with Har-SAFT consistently outperforming Geo-SAFT and Geo-SAFT outperforming Lin-SAFT. The performance metrics are detailed in **Table 2**, available at the following link: https://anonymous.4open.science/r/SAFT-MoreResult-B2D2.

---

> > ### Author Rebuttal · Reviewer_zRMC · 2026-04-03
> >
> > My scores were already aligned with the paper.

---

> > > ### Author Response · Authors · 2026-04-05
> > >
> > > Thank you for the update. We appreciate your thoughtful evaluation of our paper.

---

### Official Review · Reviewer_Z4wJ · 2026-03-06

**Soundness:** 2
**Presentation:** 2
**Significance:** 2
**Originality:** 3
**Overall Recommendation:** 4
**Confidence:** 3

**Summary:**

This work indicates that SFT imposes an excessive penalty on low-probability target tokens, while DFT suffers from vanishing gradients on these tokens, which severely hinders knowledge acquisition. To address this, the paper proposes a fine-tuning algorithm called SAFT. This algorithm specifically includes three interpolation strategies: Linear, Geometric, and Harmonic, which can adjust the gradient penalty based on the data's signal-to-noise ratio. Experiments show that this method achieves excellent performance on both mathematical reasoning and cross-domain generalization tasks.

**Compliance With Llm Reviewing Policy:**

Affirmed.

**Final Justification:**

The author's reply resolved my concerns.

**Key Questions For Authors:**

see weakness

**Limitations:**

No. The second weakness indicates that deciding which version of SAFT to use based on the performance of SFT and DFT is limited, and the author should discuss this point.

**Strengths And Weaknesses:**

**Strengths**
- The method performs well not only on in-domain mathematical reasoning tasks but also exhibits good results on OOD data such as ARC-c and MMLU-Pro.
- The experiments demonstrate good effectiveness across mainstream llms with various parameter scales and architectures.
- This work conducts mathematical research on SFT, DFT, and SAFT, providing a theoretical explanation for the effectiveness of the methods.

**Weaknesses**
- The paper dedicates a large portion to criticizing DFT for its "cold start problem", arguing that it learns hard samples too slowly. However, both Table 3 ("In-Domain Mathematical Reasoning Performance") and Table 4 ("Out-of-Domain Performance on Qwen2.5-Math-7B") show that DFT actually outperforms SFT, which contradicts the authors' motivation.
- The end of Section 4.2 states: "We establish a regime-dependent selection criterion: 1) Use Geo-SAFT for Efficiency: When the baseline SFT outperforms DFT (Signal-Dominant). 2) Use Har-SAFT for Robustness: When the baseline DFT outperforms SFT (Noise-Dominant) ." However, in real-world scenarios, we might not know whether SFT or DFT is better. Even if a benchmark exists to evaluate which one is better, it would require additional computational resources to train and evaluate both SFT and DFT models. Furthermore, even the best of the three methods only shows marginal improvements over DFT in Tables 2 and 3. The improvements for DeepSeek-Math-7B, Qwen2.5-Math-1.5B, Qwen2.5-Math-7B, and Qwen3-8B are merely 0.65, 0.7, 1.17, and 1.05 points in Table 3, respectively.
- The authors devote significant space to describing Linear-SAFT, but its actual performance is very poor.
- The hyperparameter experiment in Table 5 only tests the values 0.4, 0.5, and 0.6. It is impossible to know whether the model collapses at 0.1 or 0.9, meaning this experiment fails to demonstrate the model's true sensitivity to hyperparameters

---

> ### Author Rebuttal · Authors · 2026-03-31
>
> **1. Clarification on DFT vs. SFT Performance and Our Core Motivation**
>
> We thank the reviewer for highlighting this observation. We would like to clarify that DFT outperforming SFT in Tables 3 and 4 does not contradict our motivation; rather, it perfectly validates our core hypothesis.
>
> Our motivation is not to claim that DFT is strictly inferior to SFT, but to highlight a critical trade-off: DFT achieves noise robustness by focusing on high-confidence tokens, but pays the price of a "cold start problem" when learning hard, low-probability tokens. Mathematical reasoning datasets inherently contain noisy or suboptimal trajectories. In such noisy environments, DFT's ability to filter out outliers allows it to naturally outperform SFT.
>
> Therefore, the fact that DFT beats SFT in these tables exactly proves our premise: filtering tokens based on confidence is highly effective for noisy tasks. However, in knowledge-intensive tasks where learning from hard samples is strictly required, DFT's cold start problem severely limits its potential. This dual observation—DFT's success in noisy tasks and failure in knowledge-intensive ones—is precisely what motivated us to propose our method, which successfully resolves DFT's cold start problem while preserving its noise robustness.
>
> **2. Response to the selection criterion, computational cost, and performance improvements**
>
> We thank the reviewer for these insightful points. We address them as follows:
>
> **1) Practical Selection Criterion:** While baseline performance may be unknown a priori, task types serve as a reliable proxy. Reasoning tasks (e.g., CoT) typically have lower signal-to-noise ratios (SNR), naturally favoring DFT and Har-SAFT. Conversely, instruction-following or knowledge-intensive tasks have higher SNRs, favoring SFT and Geo-SAFT.
>
> **2) Mitigating Computational Cost:** To avoid the cost of training both SFT and DFT baselines, practitioners can evaluate on a small data subset. More importantly, our sensitivity experiments show that α=0.5 serves as a highly robust default. This allows users to achieve strong performance without prior baseline evaluation or extensive tuning.
>
> **3) Performance Improvements:** Though modest on smaller models, gains are consistent and crucially scale with model size (e.g., Qwen3-14B-base exhibits much larger gains). Furthermore, Tables 2/3 report a strict baseline using a fixed, sub-optimal α=0.5. Finer tuning yields even better results: for Qwen2.5-Math-7B, setting α=0.48 achieves 32.06% avg acc, surpassing the default 31.37%.
>
>  **3. Linear-SAFT Performance**
>
> Linear-SAFT is intentionally a baseline, as detailed in the paper (Lines 89-91, 234-236). It does not meet all requirements, which is why we introduced Har-SAFT and Geo-SAFT, which provide better functional properties.
>
> **4. Hyperparameter Sensitivity**
>
> We appreciate the reviewer’s concern. Performance curves **Figure 2,3,4,5** available at https://anonymous.4open.science/r/SAFT-MoreResult-B2D2 demonstrate the model's robustness across a wider range of hyperparameters, including extremes like 0.1 and 0.9.

---

> > ### Author Rebuttal · Reviewer_Z4wJ · 2026-04-02
> >
> > I appreciate the author's response, but unfortunately, it didn't alleviate my second concern about weaknesses. Now that we know the performance of each test set, it's easy to propose a practical selection criterion based on performance. However, what we need is a pre-existing standard for selecting a method before knowing the performance of any test sets. I've decided to maintain the score.

---

> > > ### Author Response · Authors · 2026-04-05
> > >
> > > We thank the reviewer for further clarifying this important concern. We agree that what is needed here is a **pre-existing standard** for method selection **before** observing the performance on any test set. If Sec. 4.2 were interpreted as relying on test-set performance to decide whether SFT or DFT is better, then it would indeed be a **post hoc** criterion rather than such a pre-test selection standard. We will revise the wording accordingly to make this distinction explicit.
> > >
> > > More generally, there are two natural ways to obtain a pre-test selection criterion before observing any test-set performance: 1) construct an explicit proxy metric (e.g., SNR-style measures based on consistency, confidence, or gradient stability), 2) use a standard held-out validation split for model selection. We do not view selection based on such **proxy metrics** as a strictly better practical criterion at this stage: compared with a standard held-out validation check, it requires an additional metric-design and estimation step, may incur non-trivial extra computation (e.g., repeated sampling, gradient statistics, or other estimation procedures), and often introduces a threshold-selection issue (e.g., how to define “high-SNR” versus “low-SNR”), which may vary across models and datasets. A held-out validation split, by contrast, provides a direct and standard pre-test model-selection signal, and this check can be performed on a relatively small subset of the training data, so the additional computational cost is modest. We therefore adopt **validation-based pre-test selection** as the practical protocol.
> > >
> > > Specifically, for each task, we construct **training, validation, and test splits**, train candidate methods on the training split, and use **only validation performance** to select the method before any test evaluation.
> > >
> > > First, on **Figfont**, a controlled synthetic task that approximates a model-weak regime, the model has little useful pretraining prior and must acquire a novel character-art-to-text mapping during fine-tuning. In such a setting, correct targets are often initially low-probability, so overly downweighting low-probability target tokens can hinder acquisition of the required mapping. Using **Qwen2.5-7B-base** with **40k training, 1k validation, and 5k test samples**, we obtain:
> > > | Method | Validation Acc. (%) | Held-out Test Acc. (%) |
> > > |---|---:|---:|
> > > | Geo-SAFT | 81.08 | 81.35 |
> > > | Lin-SAFT | 63.51 | 64.42 |
> > > | SFT | 39.47 | 39.21 |
> > > | Har-SAFT | 31.19 | 32.02 |
> > > | DFT | 6.88 | 6.43 |
> > > Crucially, validation already shows **SFT > DFT**; according to our guideline, this implies selecting **Geo-SAFT**, and the held-out test set confirms that **Geo-SAFT > Har-SAFT**. Thus, Figfont directly demonstrates that the rule can be applied **before seeing any test outcomes**.
> > >
> > > Second, on **NuminaMath-CoT**, this is not a new large-scale experiment added solely for rebuttal; rather, we reuse an existing completed experiment and instantiate a lightweight train/validation protocol on a subset. More importantly, in realistic fine-tuning, the purpose of training is precisely to learn useful patterns from the training distribution. In such settings, reserving a small held-out validation split from the training data to verify whether those patterns have been successfully learned—and to support checkpoint/method selection before any test evaluation—is both **reasonable and standard practice**. This also means that the decision can be made **in advance on a relatively small subset**, without requiring substantial additional computational cost. We train for **one epoch** on **6,250** training examples and use **1,250** held-out validation examples for method selection. On this validation split, **DFT = 0.3711** and **SFT = 0.2700**, indicating **DFT > SFT**; according to our guideline, this implies selecting **Har-SAFT**. On downstream benchmarks (**MATH, Minerva, Olympiad, AIME’24, AIME’25, and AMC**), we obtain:
> > > | Method | Avg. Acc. (%) |
> > > |---|---:|
> > > | Har-SAFT | 23.12 |
> > > | DFT | 22.49 |
> > > | Geo-SAFT | 21.99 |
> > > | Lin-SAFT | 16.62 |
> > > | SFT | 12.31 |
> > > The validation-based preference is therefore again consistent with downstream performance: **Har-SAFT is selected before test evaluation and indeed achieves the best overall downstream average**.
> > >
> > > Therefore, Sec. 4.2 should be understood not as using test performance to choose a method, but as a regime-dependent preference that can be implemented via held-out validation performance. In realistic fine-tuning, using a small validation split for method selection before any test evaluation is natural and standard practice. This protocol is already strictly pre-test, stable, inexpensive, and easy to reproduce. We will revise the section to make this distinction explicit.

---

### Official Review · Reviewer_qtwm · 2026-03-16

**Soundness:** 2
**Presentation:** 3
**Significance:** 3
**Originality:** 2
**Overall Recommendation:** 4
**Confidence:** 3

**Summary:**

This paper proposes Spectrum-Adaptive Fine-Tuning (SAFT), a family of token-reweighted SFT objectives that interpolates between standard NLL/SFT and DFT using linear, geometric, and harmonic weighting schemes. The method reweights token losses based on predicted probabilities to adjust the gradient contributions of high- vs low-confidence tokens. Empirically, the paper reports that Har-SAFT consistently performs best across five model backbones and six math benchmarks, slightly outperforming DFT on in-domain averages. Additional experiments show improved robustness to synthetic label noise and improved out-of-domain (OOD) performance on several reasoning benchmarks.

**Compliance With Llm Reviewing Policy:**

Affirmed.

**Final Justification:**

I have raised a few questions regarding comparison with baselines, evaluation setups, hyperparameter settings. The authors have conducted the additional experiments which resolves several of the concerns.

As a result, the rating is updated from 3 to 4 after the first rebuttal.

**Key Questions For Authors:**

- How consistent are these improvements across multiple random seeds and training runs?

**Limitations:**

yes

**Strengths And Weaknesses:**

**Strengths**
- Experiments cover multiple backbones (5 models) and six math datasets, which gives a reasonably broad empirical picture.
- The method is simple to implement and introduces only a small modification to the standard SFT objective.
- The paper provides a smooth interpolation between SFT and DFT, which helps conceptualize how probability-based reweighting affects optimization.

**Weaknesses**
- There are several related paper not discussed here, it is important for the authors to compare the contribution relative to recent work on SFT objective design, such as [1,2]:

[1] On the Generalization of SFT: A Reinforcement Learning Perspective with Reward Rectification (https://openreview.net/forum?id=Lv7PjbcaMi)
[2] Proximal Supervised Fine-Tuning (https://openreview.net/forum?id=hQtwQqYikp)

The paper would benefit from clearer positioning relative to these approaches, which also modify the SFT objective to improve generalization.

- The OOD improvement is largely driven by large gains on ARC-c, where the base model performance appears unusually low. I would suggest the authors to provide more discussion and carefully revisit the evaluation setup.

- Hyperparameter analysis is minimal (only Har-SAFT, single model, narrow \alpha range). It would strengthen the paper to demonstrate robustness across \alpha values and models.

---

> ### Author Rebuttal · Authors · 2026-03-31
>
> **1. Comparison with Related Works**
>
> Regarding DFT [1]: DFT serves as the foundation for our SFT-DFT interpolation. Our method differs by interpolating between SFT and DFT, balancing their strengths and weaknesses. **DFT is already discussed and compared in detail in our paper as baseline**.
>
> Regarding PSFT [2]: First, PSFT, based on the PPO algorithm, controls model updates using clipping to suppress overconfidence. In contrast, our method fundamentally differs by employing a soft dynamic weighting mechanism based on the model's absolute confidence, rather than a hard trust-region clipping based on relative policy changes, allowing for a smooth interpolation between SFT and DFT. Second, we benchmark the official PSFT code on our datasets and observed a poor performance-efficiency trade-off, producing significantly worse results than SFT with the same number of epochs. To investigate if PSFT simply requires more convergence time, we extend its training by two additional epochs on Qwen2.5-Math-1.5B, confirming that its **conservative update strategy severely delays convergence**. In contrast, our method improves training efficiency on complex datasets by applying interpolation strategies that ensure both robustness and efficiency. More results on PSFT at **Table 1** and the extended performance curve are available in **Figure 1** of the following link: https://anonymous.4open.science/r/SAFT-MoreResult-B2D2.
> |        Metric       | DeepSeek-7B | Qwen2.5-7B | Qwen2.5-1.5B | Qwen3-14B | Qwen3-8B |
> | ------------- | ----------- | ---------- | ------------ | --------- | -------- |
> | Avg. Acc. (%) | 7.6         | 10.3       | **10.9**     | 8.9       | 9.0      |
>
> **2. OOD Improvement and Evaluation Setups**
>
> We emphasize that our method yields consistent OOD improvements across multiple benchmarks, including gains on **MMLU-Pro** and **GPQA***. Alongside these broad improvements, we observe particularly substantial gains on ARC-c. Regarding ARC-c, while the base model performance might initially appear low, this is consistent with established baselines in prior work. Upon reviewing Luffy’s study, we found their base model performance was also low (18.2%) [3]. The **only difference** in our evaluation setup is the **maximum inference length setting (4k vs. 8k)**, which accounts for the slight variance in exact scores. Specifically, our base model, with max_tokens set to 8192, achieved **ARC-c = 16.81, GPQA\* = 12.12, and MMLU-Pro = 18.26**, demonstrating strong consistency with Luffy's evaluation.
>
> Upon analyzing the base model’s outputs, we find that many responses marked as incorrect are actually due to formatting issues. After correcting misclassified answers due to format issues, the base model scored 48.6%, **still much lower than the Hybrid model at 70.39%**, highlighting the clear performance gap and the improvement in reasoning ability.
>
> **3. Hyperparameter Analysis**
>
> We extend our hyperparameter analysis to demonstrate robustness across a wider range of settings. Specifically, we plot performance trends for Har-SAFT on Qwen2.5-Math-1.5B and Qwen2.5-Math-7B as α varied from 0.1 to 0.9, along with performance for the three different interpolation methods on Qwen2.5-Math-1.5B. The results show consistent improvements across different \alpha values, confirming the robustness of our method. The performance curves can be found at **Figure 2,3,4,5** in the following link: https://anonymous.4open.science/r/SAFT-MoreResult-B2D2.
>
> **4. Consistency Across Random Seeds and Multiple Runs**
>
> We conduct training with **two different random seeds** on Qwen2.5-Math-1.5B and perform **three independent runs** for each seed. As shown in **Figure 6** at https://anonymous.4open.science/r/SAFT-MoreResult-B2D2, our method yields stable and consistent improvements across different seeds and runs, supporting the robustness of our findings.
>
> *Reference*
>
>  [1] DFT, ICLR 2026.
>
>  [2] PSFT, ICLR 2026.
>
>  [3] Luffy, NeurIPS 2025.

---

> > ### Author Rebuttal · Reviewer_qtwm · 2026-04-03
> >
> > The authors have conducted additional analysis and checked evaluation settings. I have hence increased my rating.

---

> > > ### Author Response · Authors · 2026-04-05
> > >
> > > Thank you for reassessing the rating. We appreciate your insightful suggestions and will make appropriate revisions to our paper.

---

### Decision · Program_Chairs · 2026-04-30

**Decision:**

Accept (regular)

**Comment:**

This paper proposes SAFT, a family of post-training objectives interpolating between SFT and DFT. Reviewers were overall positive after the rebuttal and converged to weak accept, citing the simplicity of the method, broad experimental coverage, and improved robustness. The rebuttal also addressed the main technical issue by correcting the original gradient-based framing. The main remaining concerns are that the novelty should be positioned more modestly relative to recent work on alternative supervised post-training objectives, that the practical choice between SAFT variants remains somewhat heuristic, and that some of the empirical claims should be stated more conservatively.